# Review on Immersion Vaccines for Fish: An Update 2019

**DOI:** 10.3390/microorganisms7120627

**Published:** 2019-11-29

**Authors:** Jarl Bøgwald, Roy A. Dalmo

**Affiliations:** Norwegian College of Fishery Science, Faculty of Biosciences, Fisheries and Economy, University of Tromsø—The Arctic University of Norway, N-9017 Tromsø, Norway; jarl.bogwald@uit.no

**Keywords:** immersion vaccination, vaccines, fish, diseases, aquaculture

## Abstract

Immersion vaccines are used for a variety of aquacultured fish to protect against infectious diseases caused by bacteria and viruses. During immersion vaccination the antigens are taken up by the skin, gills or gut and processed by the immune system, where the resulting response may lead to protection. The lack of classical secondary responses following repeated immersion vaccination may partly be explained by the limited uptake of antigens by immersion compared to injection. Administration of vaccines depends on the size of the fish. In most cases, immersion vaccination is inferior to injection vaccination with regard to achieved protection. However, injection is problematic in small fish, and fry as small as 0.5 gram may be immersion vaccinated when they are considered adaptively immunocompetent. Inactivated vaccines are, in many cases, weakly immunogenic, resulting in low protection after immersion vaccination. Therefore, during recent years, several studies have focused on different ways to augment the efficacy of these vaccines. Examples are booster vaccination, administration of immunostimulants/adjuvants, pretreatment with low frequency ultrasound, use of live attenuated and DNA vaccines, preincubation in hyperosmotic solutions, percutaneous application of a multiple puncture instrument and application of more suitable inactivation chemicals. Electrostatic coating with positively charged chitosan to obtain mucoadhesive vaccines and a more efficient delivery of inactivated vaccines has also been successful.

## 1. Introduction

Disease prevention by vaccination is, on economic, environmental and ethical grounds, the most appropriate method for pathogen control currently available to the aquaculture industry. Treatment of fish diseases with antimicrobials may have negative impacts on the aquatic environment and human health. Traditionally, vaccines comprise either live, attenuated, replicating or non-replicating pathogens, inactivated pathogens or their subunits. Inactivated vaccines based on either killed pathogens or isolated pathogen subunits are, in many cases, weakly immunogenic with low vaccine efficacies. Immersion vaccination is more applicable compared to injection vaccination, but the method suffers from a low potency, due primarily to inefficient uptake of antigens across mucosal membranes. 

Immersion vaccination involves immersion of fish in water containing vaccine antigens. Dip vaccination is rapid, as the fish are immersed in water containing a relatively high dose of vaccine antigen(s) for one or several minutes, or, if bath vaccinated, the fish receive a more diluted vaccine antigen preparation for a longer period. Fish can be booster vaccinated by dip or bath in order to increase protection. Immersion vaccination is suitable for mass vaccination of fish too small for high throughput injection vaccination. Unfortunately, the vaccine efficacies displayed from immersion vaccines are low to moderate in most instances, even though many exceptions exist [1]. It is quite difficult to pinpoint why some vaccines show high efficacies and some show low efficacies. Many variables for vaccine efficacy are present and should be considered when conducting immersion vaccination trials [2]. These include vaccine (antigen) dose, duration of immersion, particulate/soluble antigen uptake during immersion immunization of fish [3], adjuvant performance [4,5,6,7,8,9], temperature [10], fish size (age) [10,11], osmolarity, prime boost strategy [1], mucosal integrity [12,13], replicative vs. non-replicative vaccines [1] and how the experimental pathogen challenges are carried out (e.g., virulence of the challenge pathogen, high or low pathogen pressure/load). 

Both optimal duration and vaccine dose during immersion are vital to achieve a good vaccine response, as reported by Du et al (2018) [14]. To increase immersion vaccine efficacy, several new methods have been developed. By using hyperosmotic environment, as reported by Huising et al (2003) [15], and later by Gao et al. 2016 [16], vaccine efficacy can be increased compared to traditional methods that involve the administration of inactivated vaccine antigens by bathing. Furthermore, other modalities to increase antigen uptake during immersion vaccination have been developed. The first one described increased the adhesion and uptake of antigens of inactivated Flavobacterium when coated by positively charged chitosan which displayed mucoadhesive properties. This modality increased vaccine efficacy compared to what was obtained using naked vaccine antigens [17]. The other one used TNF alpha (TNF-α) nanoparticles which hold promise as an adjuvant for immersion vaccination [4]. Further on, recent studies suggest that nanoliposomes [18], recombinant live viruses expressing protective antigens, attenuated live vaccines [19,20,21,22] and microbubbles [13] may be used to increase the vaccine efficacy of immersion vaccines. In fact, live attenuated immersion vaccines have been commercialized for catfish or salmonids in USA/Canada (AQUAVAC-ESC^®^, AQUASOL-COL^®^, Renogen^®^), reviewed by Shoemaker et al. (2009) [23]. Many of these new methods are described in this review, in relation to how they have been applied to a fish species—especially where vaccine efficacy determination has been carried out.

It is important to differentiate immersion vaccination from oral vaccination. Usually, inactivated heat or formalin treated pathogens +/- extracellular products are used for immersion vaccination, while feed encapsulated antigens are used for oral vaccination. The initial uptake and processing of antigens occur exclusively in the gut after oral vaccination, whereas several tissues and cells (including the gut) may take up and process antigens during immersion vaccination. Antigen degradation, due to pH (stomach) and enzymatic breakdown (intestine), is a major issue when developing oral vaccines. Antigens may, however, be protected from breakdown by polymer vehicles (alginate, chitosan, liposomes, PLGA particles) which may improve vaccine effects, reviewed by Ji et al. 2015, Mutoloki et al. 2015, and Embregts and Forlenza 2016 [24,25,26]. Interestingly, the use of certain adjuvants (e.g., rTNFα) may counteract poor vaccine efficacy [27]. An optimal formulation for oral vaccines has not yet been found; this may be due to differences between fish species, dose–effect relationship, ontogeny of immune system, induction of tolerance, whether the inactivated pathogen/antigen display immunogenic determinants, etc. Lastly, the vaccine must be cost-efficient.

## 2. Mucosal Immune Response 

It is acknowledged that the mucosal lining is of high importance when preventing pathogen entrance. Immune cells are present in all mucosal tissues/linings (MALT), from nasopharynx-associated lymphoid tissue (NALT), gill-associated lymphoid tissue (GIALT), skin-associated lymphoid tissue (SALT), buccal cavity-associated lymphoid tissue, and gut-associated lymphoid tissue (GALT). These tissues contain characteristics important to the adaptive immune system [28,29,30,31,32,33]. The mucosal integument and soft tissues also contain innate defense molecules which may be fully protective against early stage infection [34]. These innate molecules may also be modulated by immersion treatment, feeding or by infection [20,34,35,36,37,38,39,40,41,42,43,44,45,46,47,48,49,50,51,52,53,54].

Strictly speaking, SALT and GALT do not contain well-organized lymphoid assemblages, but a more diffuse occurrence of immune cells—so-called diffuse mucosa-associated lymphoid tissue (D-MALT). Nevertheless, all ALTs may possess anti-infectious and pro-inflammatory elements important for protection, as they are innate lymphocyte-derived mechanisms or immunoglobulins (in addition to innate defense factors). After immersion vaccination, there is a so-called disparity when it comes to antibody responses: mucosal immunization may induce localized mucosal immune responses, whereas parenteral administration induces systemic response and production of specific antibodies [52,55,56]. It is difficult to pinpoint which MALT tissue(s) are most important when it comes to immune response and protection elicited after immersion vaccination. Likely, there are inter-species differences as well as differences between different mucosal sites, as pointed out by Khansari et al., 2018 [52]. IgT (IgZ in zebra fish) has been proposed to be the central immunoglobulin produced at mucosal sites following exposure to parasites [57]. Furthermore, IgT expression was shown to be highly tissue-dependent after immersion vaccination against rainbow trout fry syndrome. In the latter study, IgT was temporarily increased in the posterior intestine while down-regulated in the gills and skin [58]. However, IgM is also present at mucosal sites after mucosal immunization, but is not as regulated as IgT. The relative importance of IgT response compared to IgM is debated. Inter-species differences, mode of vaccine administration, and stimulation/immunization/infection may be decisive for differential IgT and IgM responses [59]. As well as this, not every teleost species possesses the IgT isotype [60]. It might appear that IgM is the only consistent Ig isotype at mucosal sites in all teleost species. IgT response has been scarcely assessed after the immersion vaccination in previous scholarly work. One reason for this is the lack of specific antibodies against IgT from various fish, which excludes immunohistochemical analysis and ELISA based assays. Local induction and production of both mucosal (IgT) and systemic (IgM) may be present upon microbial exposure [61], and in granulomatous tissues induced by the intraperitoneal and intramuscular injection of vaccines [62,63]. Besides mucosal IgT, secretory polymeric Ig receptors (pIgR) with the ability to bind IgM and IgT has been shown to transport IgM and IgT to the gut luminal area. A recent study showed increased skin pIgR and IgM mRNA expression following immersion vaccination against *Vibrio anguillarum* in flounder (*Paralichthys olivaceus*). This finding was confirmed using ELISA and immunohistochemistry [64]. For more information on mucosal immunoglobulins and pIgR, see recent updates [60,65,66,67,68,69]. The expression of Ig and pIgR at mucosal sites is of vital importance with respect to mucosal binding and opsonization of IgM and IgT to antigens/pathogens. However, one should not underscore the importance of innate defense factors after immersion vaccination, which may also contribute to cross-protection [70]. This topic will not be addressed further in the present review.

One peculiarity is extracellular vesicles (EV) in fish mucus, which are probably products from skin cells. The EV obtained from Atlantic cod mucus contains complement component C3, which is the central component in the alternative and classical pathway of complement activation. The cod EV also contains C-reactive protein, histone H3, galectins and profilin, in addition to other proteins [71]. Given that there are numerous other (immune) proteins in the EVs, they should be given further attention, with regards to their induction and release during immersion vaccination [72].

## 3. Atlantic Cod 

### Vibriosis

Farming of Atlantic cod (*Gadus morhua* L.) was a growing industry in Norway (Ca 21,000 tons sold in 2010), but production decreased to a negligible level in 2015–2018. In Atlantic cod aquaculture, classical vibriosis is the most significant bacterial disease [73]. Intensively reared juveniles of approx. 1 g were continuously fed in seawater of 3.4% salinity at 10 °C, and dip vaccinated with a commercial vaccine (ALPHA MARINE^TM^ Vibrio) with *V. anguillarum* serotype O1, O2a and O2b [73,74]. The fish were highly protected against vibriosis when bath challenged with both serogroup O2a and O2b, but the vaccine protected poorly against a deviating serotype O2 isolate. In a more recent study, Atlantic cod juveniles of approx. 3.7 g were dip vaccinated for 30 sec at 10 °C [73]. Three monovalent and one trivalent experimental dip vaccine were produced (ALPHARMA AS (Norway)) containing bacterins of Va-O2a, Va-O2b and Va-O2. The trivalent vaccine included all the isolates Va-O2a, Va-O2b and Va-O2. Vaccine efficacies were determined 7 weeks post-vaccination by bath challenges with Va O2a, Va-O2b and the deviating Va-O2. The monovalent vaccines were highly protective, resulting in a relative percent survival (RPS) of 93, 87 and 86 against homologous challenge with Va-O2a, Va-O2b and Va-O2, respectively. The trivalent vaccine protected efficiently against all serogroups tested, with an RPS above 90 [73]. Immersion vaccination has not been tried just for Atlantic cod, but for many other fish species. Table 1 gives an overview of selected immersion vaccination trials where experimental vaccines have been examined for protection against challenge pathogens.

## 4. Atlantic Salmon/Chinook Salmon/Coho Salmon Yersiniosis (Enteric Redmouth Disease) and Flavobacteriosis

Midtlyng and Bravo (2007) have previously summarized the availability of fish vaccines in Chile in the period 1999–2003 [91]. Chilean salmon farming started in the early 1980s and has been one of the most successful sectors of the country’s economy. Production mainly takes place in the south and is dominated by the marine culture of Atlantic salmon (*Salmo salar*), coho salmon (*Oncorhynchus kisutch*) and rainbow trout (*Oncorhynchus mykiss*). In 2005, Chile was the world’s largest salmonid fish producer, with about 585,000 tons harvested. In Chile, as in other countries, infectious diseases are among the most serious problems of aquaculture. Salmonid Rickettsial Septicemia (SRS) caused by the intracellular bacterium *Piscirickettsia salmonis* has been considered the main cause of mortality in farmed coho salmon since the beginning of the industry [91]. The vaccines applied by immersion comprise 29% of the total sales volume, 85.2% of which vaccines against yersiniosis, followed by bivalent (*Flavobacterium columnare*/yersiniosis) vaccines, with 9.7%. Vaccines against yersiniosis and flavobacteriosis are normally administered by immersion before transfer of the salmon fry to the freshwater lakes. The fish were immunized once at the size of 1 g and boosted by immersion again when slightly above 5 g. Vaccines against yersiniosis have proven their efficacy as there are no, or only scarce, outbreaks. This has contributed to minimizing the use of antibacterial drugs in freshwater lakes. Unlike other bacterial diseases, vaccination against SRS has apparently not significantly reduced the need for treatment, as the amount of antibiotics used for control remained at the same level.

Generally, salmonid fry are initially immunized with *Y. ruckeri* by immersion vaccination at the size of 2–5 g. Before reaching this size, *S. salar* fry are not considered to have developed sufficient adaptive immunocompetence. The risk of infection in small fish by *Y. ruckeri,* due to the ubiquity of the bacterium, and its ability to survive in the environment without a host, is considerable. In one study, Atlantic salmon fry of mean weight 0.26 g were dip vaccinated with inactivated *Y. ruckeri* for 60 sec (one dip vaccine) before being returned to their respective tanks [83]. Another group of fry was dip vaccinated twice with a booster vaccination at the mean size of 1.2 g (two dip vaccinations). Nine weeks after booster vaccination the fry were challenged by a 60 min immersion in *Y. ruckeri* in fresh water at a final concentration of 2.5 × 10^7^ colony forming units (CFU mL^−1^). Tanks were monitored for mortalities and sampled for 21 days post-challenge. The protection was low in both groups, with an RPS of 20.4 and 16.7 for 1 dip and 2 dip vaccine groups, respectively [83].

Enteric redmouth disease and yersiniosis are closely related fish diseases. Both cause bacterial hemorrhagic septicemia and are caused by the bacterium *Y. ruckeri*. ERM was first reported in the 1950s and has since been reported throughout all trout farming regions in the Northern hemisphere, and significantly impacts the culture of rainbow trout. In another study, Atlantic salmon (*S. salar*) weighing 5 g were both vaccinated and challenged via immersion. The fish were vaccinated by a trypsinized version of yersinivac-B (MSD Animal Health, Australia) and maintained in fresh water at 15 °C. At six weeks post-vaccination the fish were challenged with *Y. ruckeri* O1b, 4.3 × 10^6^ colony forming units mL^−1^. Cumulative mortality 21 days post-challenge in the unvaccinated control group plateaued at 83%, and the relative percent survival (RPS) was calculated to 57% [82].

## 5. Rainbow Trout

The annual world-wide production of rainbow trout (*O. mykiss*) has declined in recent years, from 883 kilotons in 2012 to 814 kilotons in 2016. Chile, Norway, France, UK, Italy, Spain and other European countries are the main producers. Rainbow trout may suffer from a number of diseases caused by viruses and bacteria.

### 5.1. ERM in Rainbow Trout

Numerous commercial immersion vaccines against ERM exist, mainly based on serotype O1, biotype 1. The vaccination involves 30 sec of immersion of rainbow trout fry into a diluted formalin inactivated vaccine (bacterin) and offers protection for a limited period of time. Booster vaccination is needed to achieve complete protection for the rest of the production cycle. The vaccine may not give protection against antigenic variants, e.g., biotype 2, as observed in biotype 1 vaccinated fish. The present work compares protection 4- and 6 months post-vaccination with vaccines based on either biotype 1 or both biotypes 1 and 2 [85]. Immersion vaccination of 5 g juvenile rainbow trout took place 3.5 months post-hatch in a diluted bacterin (1:10) for 30 s. For challenge experiments, 6–7 × 10^7^ CFU per fish were intraperitoneally injected. Lower survival rates (RPS) were found in biotype 1 vaccinated groups, (13.5%) and (29.5%), compared to a combination of both biotype 1 and 2 vaccinated groups, (42.5%) and (52%), respectively. These results indicate a significant increase in the survival rates of fish vaccinated with both biotype 1 and biotype 2, compared to only biotype 1 vaccinated and unvaccinated fish.

A single immersion vaccination of rainbow trout fry using *Y. ruckeri* bacterin confers immunity to reinfection, but only for a shorter period. This article shows that booster vaccination prolongs immunity. A booster vaccination using dilutions of the bacterin (1:100, 1:1000, 1:2000) with increased exposure time (1 h, 2 h) confers a higher and longer-lasting immunity, although a short term (30 s) booster in 1:10 diluted bacterin proved superior [84]. In this study, rainbow trout of 5 g were immersion vaccinated with a diluted vaccine (1:10) of formalin inactivated *Y. ruckeri* biotype 1 (5 × 10^9^ cells mL^−1^) for 30 s. The primary vaccination was followed by booster vaccination after 1 or 2 months in a diluted vaccine. Challenge was performed 2- and 4 months post-boost. In the first challenge experiment (4 months post-booster, 5 months post-primary vaccination), the lowest mortality was recorded in fish vaccinated 2× in 1:10 diluted vaccine, namely 24%. Fish that had been booster vaccinated in a diluted bacterin for 1 h showed a lower mortality, compared to fish vaccinated only 1×. In the second challenge experiment, 2 months post-booster, 4 months post-primary vaccination the lowest mortality was recorded in the vaccinated 2× (1:10) group as 34%. Groups which had been subjected to a diluted vaccine (1:100, 1:1000, 1:2000) showed a mortality of 60%–62%, whereas the unvaccinated fish and the fish vaccinated only once showed mortalities of 88% and 84%, respectively [84].

In another study, rainbow trout with a mean weight of 6.3 g were immersion vaccinated with purified lipopolysaccharide (LPS) from the fish pathogen *Y. ruckeri* once and twice (interval of 20 days booster). The fish were challenged by intraperitoneal injection of 9.8 × 10^6^ virulent *Y. ruckeri* per fish. 28 and 60 days after the last immunization. A significantly lower mortality was achieved in all vaccinated groups—both the prime vaccinated and boosted groups—compared with nonvaccinated controls. The results indicate that *Y. ruckeri* LPS is a protective immunogenic compound in rainbow trout [90].

This following study reports how rainbow trout respond to repeated immersion vaccination against yersiniosis (ERM) caused by the bacterial pathogen *Y. ruckeri* [86]. It was found that rainbow trout do not raise a classical secondary response following repeated immersion vaccination. Fish were vaccinated by a 30 s immersion in a bacterin diluted 1:10 containing *Y. ruckeri* serotype O1, biotype 1 and biotype 2 (5 × 10^9^ cells mL^−1^ each). Sublethal exposure of vaccinated fish to live bacteria was performed with a Danish strain of *Y. ruckeri* serotype O1, biotype 2. Specific *Y. ruckeri* antibody (IgM) levels in vaccinated and unvaccinated fish did not differ significantly before challenge, but a significant increase in the antibody level was seen in all groups, 3 weeks post-exposure to live bacteria. The densities of IgM-positive lymphocytes in the spleen of three times immunized fish increased compared to control fish, but no general trend for an increase with number of immunizations was noted. The lack of classical secondary responses following repeated immersion vaccination may partly be explained by the limited uptake of antigens by immersion compared to injection [86].

Vaccination of rainbow trout against enteric redmouth disease (ERM) by immersion in *Y. ruckeri* bacterin confers a high degree of protection to the fish. The present study demonstrates a significant increase in plasma antibody titers following immersion vaccination and significantly reduced mortality during *Y. ruckeri* challenge. Rainbow trout were immersion vaccinated with a *Y. ruckeri* bacterin. The fish were subsequently exposed to 1 × 10^9^ CFU *Y. ruckeri* mL^−1^ for 1 h, either 8 or 26 weeks post-vaccination [87]. All vaccinated groups showed 0% mortality when challenged, compared to non-vaccinated controls (40% and 28% mortality after 8 and 26 weeks). A significant increase in specific IgM antibody levels was recorded in vaccinated fish, which also showed a reduced bacteremia during challenge. In vitro plasma studies showed a significantly increased bactericidal effect of fresh plasma from vaccinated fish, indicating that plasma proteins may play a role in the protection of vaccinated rainbow trout. 

Immunostimulants activate both innate and adaptive immune responses. The present study reports the influence of intraperitoneally injected dimerized lysozyme (KLP-602) on the cellular and humoral immune response after the immersion immunization of rainbow trout with *Y. ruckeri* vaccine. Dimerized lysozyme administered before or along with the vaccine induced higher levels of specific antibody-secreting cells and specific humoral immune response, compared with fish in the control group [88].

Potentially immunostimulatory effects of orally administered ß-glucan were investigated in combination with immersion vaccination against enteric redmouth disease caused by *Y. ruckeri* in rainbow trout (*O. mykiss*). A ß-glucan was applied at an inclusion level of 1% in feed administered at a rate of 1% biomass day^−1^ for 84 consecutive days. Fish were vaccinated after two weeks of experimental feeding and bath-challenged with live *Y. ruckeri* six weeks post-vaccination. A cumulative mortality of 16.7% was observed in unvaccinated controls post-challenge and vaccinated fish experienced significantly increased survival (RPS = 85%). The ß-glucan had no effect on survival in either unvaccinated or vaccinated fish [89]. 

Immersion vaccination of rainbow trout against *Y. ruckeri* infection is an established method to prevent ERM, but the effect is inferior to injection vaccination, and the duration of protection is limited to less than six months. The work by Skov et al. (2018) [6] shows how ERM immersion vaccination of rainbow trout, in combination with exposure to soluble adjuvants, Montanide IMS1312 VG PR and ß-glucan, affects immune reactions. In this study, fish were exposed to either 1) a 30 s dip in a 1:10 dilution of the vaccine Aquavac Relera Vet, 2) a 30 s dip in vaccine + montanide, 3) a 30 s dip in vaccine + ß-glucan, 4) a 30 s dip in Montanide alone, and 5) a 30 s dip in ß-glucan alone and unvaccinated controls. Thirty and 42 days after vaccination, fish were bath-challenged with a primary challenge performed for 6 h in a concentration of 1.4 × 10^7^ CFUml^−1^ of *Y. ruckeri* O1 biotype 2, and a secondary challenge for 8 h in 1.1 × 10^7^ CFU mL^−1^ of the same strain [6]. Montanide, when used alone, induced a slightly increased, but not statistically significant protection, whereas ß-glucan did not increase protection. Specific antibody production was not positively affected by combining adjuvant and vaccine. Mucosal immune response genes were upregulated 24 h post-vaccination in fish gills exposed to both vaccine-adjuvant combinations when compared to fish exposed to vaccine alone [6].

### 5.2. Vibriosis

For immersion vaccination, rainbow trout were held in a net for 30 s in a diluted inactivated suspension of serotype O1. Immunized fish were challenged with intraperitoneal injection of a *V. anguillarum* virulent *V. anguillarum* O1 strain. The immersion vaccination provided a high level of protection, with an RPS of 93 and 81.8, for two- and three-months post-vaccination, respectively [79].

### 5.3. Viral Hemorrhagic Septicemia (VHS)

Whether an infection with a pathogenic virus in fish results in the development of clinical disease often depends on the balance between virus multiplication and immune reactions in the host. Low temperatures have been reported to delay the adaptive immune response in salmonids. A DNA vaccine encoding the viral glycoprotein G was found to protect rainbow trout efficiently against viral hemorrhagic septicemia (VHS) at 5, 10 and 15 °C, suggesting that long lasting innate mechanisms compensated for the slow development of adaptive immunity at low temperatures [110].

The DNA-immersion vaccination of fingerling rainbow trout was performed by the use of a plasmid coding for the glycoprotein G gene of the viral hemorrhagic septicemia virus (VHSV). Short pulses of low intensity ultrasound were the only method by which both humoral antibody responses and survival after VHSV challenge were obtained. Fingerling trout of 5–7 g were immersed in a 10 µg mL^−1^ G3-pcDNAI/Amp plasmid, including 24 s with ultrasound. One month after vaccination, the fish were challenged by intraperitoneal injection of 10^6^–10^7^ plaque-forming units of VHSV per trout. The highest anti-VHSV G antibody titers were obtained in sera from trout immunized in G3-pcDNAI/Amp plasmid with ultrasound, and a statistically significant higher rate of survival (RPS of 50.1) than trout not exposed to plasmid 38 days after challenge [109].

### 5.4. Rainbow Trout Fry Syndrome/Bacterial Cold-Water Disease (RTFS/BCWD)

*Flavobacterium psychrophilum*, the causative agent of RTFS and BCWD, causes high mortality among hatchery-reared rainbow trout in Europe and the USA. No commercial vaccines have yet been developed. One obstacle is that fry that are less than 0.5 g are susceptible to RTFS and are not considered fully immunocompetent, as suggested by Salinan et al. (2015) [119]. However, a number of studies have shown that immersion immunization can be successful in fish of 0.5 g or larger. Consequently, only oral or bath vaccines are relevant. The following study investigates whether what was presumably the most potent immersion immunization was able to induce immunity to a subsequent intraperitoneal challenge of non-attenuated isolates of *F. psychrophilum*. Rainbow trout of 1 g were immersed in a live *F. psychrophilum* suspension of 2.0 × 10^8^ CFU mL^−1^ for 30 min, and rainbow trout of 1.3 g were immersed for 50 min. For comparison, rainbow trout were immersed in a suspension of formalin-killed cells. Twenty-six days post-immersion the fry were intraperitoneally injected with 1.6 × 10^8^ CFU, and 47 days post-immersion another set of fish were intraperitoneally injected with 2.9 × 10^7^ CFU *F. psychrophilum*. Immersion in live bacteria for 30 or 50 min caused no mortality and protected a major fraction of the fry against challenges 26 and 47 days later, with RPS values of 88.2% and 60.3%, respectively [97]. Increased specific antibody titers suggested that adaptive immune mechanisms were involved in the protection. 

BCWD caused by *F. psychrophilum* remains one of the most significant bacterial diseases of salmonids worldwide. A previously developed and reported live-attenuated immersion vaccine (*F. psychrophilum*; B.17-ILM) has been shown to confer significant protection in salmonids. To further characterize this vaccine, experiments were carried out to determine the cross-protective efficacy of this B.17-ILM vaccine against nine *F. psychrophilum* isolates (representing seven sequence types/three clonal complexes, as determined by multi-locus sequence typing) in comparison with a wild-type virulent strain, CSF-259-93 [96]. To assess protection, experimental challenges of rainbow trout fry were conducted following immersion vaccinations with the B.17-ILM vaccine. All *F. psychrophilum* strains used in challenge trials were isolated from several fish species, and all were found to be virulent in rainbow trout. Juvenile rainbow trout of 2.5 g were immersed for 3 min in a B.17-ILM vaccine of 10^10^ CFU ml^-1^ diluted 1:10. Two weeks following initial vaccination, all fish were booster vaccinated in an identical manner to the primary vaccination. Eight weeks following initial vaccination, fish were challenged with 10 different *F. psychrophilum* strains by intramuscular injection. Cumulative percent mortality was recorded 28 days after challenge, and relative percent survival (RPS) was calculated to 51%–72%. This study demonstrates clearly that the B.17-ILM vaccine confers solid cross-protection in rainbow trout and represents a promising vaccine for global aquaculture.

A live attenuated immersion vaccine (B.17-ILM) against bacterial cold-water disease (BCWD) caused by *F. psychrophilum* in salmonids has recently been developed. Optimization of efficacy of the vaccine was performed in rainbow trout fry by investigating fish size, vaccine delivery time, dose, booster regimes and duration of protection [95]. Immersion vaccination for 3, 6 and 30 min produced significant protection with RPS values of 47, 53 and 52, respectively. The vaccine provided significant protection for fish as small as 0.5 g (RPS 55%), 1 g (RPS 59%) and 2 g (RPS 60%). Fish vaccinated with higher doses of 10^10^ and 10^8^ CFU mL^−1^ were strongly protected for at least 24 weeks, with RPS values up to 70%. Fish vaccinated with lower doses 10^6^ and 10^5^ CFU mL^−1^ had good protection for 12 weeks, but RPS values dropped to 36 and 34, respectively, by 24 weeks. Vaccine efficacy was optimum when the primary vaccination was followed by a single booster (RPS 61%) rather than two boosters (RPS 48%). Vaccination without a booster resulted in a lower RPS (13%). All vaccinated fish developed significantly higher serum antibody levels by week 8 compared to their respective controls [95].

### 5.5. Streptococcosis

A new administration method was developed for the vaccination of juvenile rainbow trout against ß-hemolytic *Streptococcus*. Small skin lesions were produced using a multiple puncture instrument while fish were immunized in a vaccine suspension containing formalin-killed *Streptococcus iniae* 10-fold diluted vaccine of 10^11^ CFU mL^−1^. Two weeks after immunization, fish were challenged by intraperitoneal injection of live *S. iniae* at doses of 2.3 × 10^2^, 2.3 × 10^3^ or 2.3 × 10^4^ CFU per fish in the first experiment, and 2.8 × 10^2^, 2.8 × 10^3^ or 2.8 × 10^4^ CFU per fish in the second experiment. The mortality of fish vaccinated by this method was 40%, equal to that by intraperitoneal vaccination, while non-vaccinated control fish and fish vaccinated by immersion without multiple puncture each experienced 80% mortality [12]. Quantitative analysis using fluorescent microspheres revealed that both antigen uptake by skin and delivery to the kidney and spleen were more effective with this method compared with the immersion alone.

### 5.6. Furunculosis

*Aeromonas salmonicida* is a non-motile, facultatively anaerobic, Gram-negative bacterium. It is the etiological agent of salmonid furunculosis, a disease capable of causing serious losses in both cultured and wild stocks of salmonids. To date, injection vaccination is the only route of administration which has provided reasonable levels of protection. This study reports a slow-growing, aminoglycoside-resistant mutant, and a rapidly-growing pseudo-revertant, of *A. salmonicida*. These mutants differed morphologically from the wild-type and from one another with respect to A-layer organization, membrane antagonist sensitivity and aerobic metabolism. Both mutants were avirulent and incapable of tissue persistence. The rapidly-growing, antibiotic-resistant pseudo-revertant, when administered either intraperitoneally or by immersion, effectively protected salmonid fish from challenge by a heterologous virulent strain [100]. 

## 6. Ayu

### Bacterial Cold-Water Disease (BCWD)

Ayu (*Plecoglossus altivelis*) is one of the most important fish species for freshwater fisheries in Japan. *F. psychrophilum* is the causative agent of bacterial cold-water disease (BCWD) that occurs in ayu, and is one of the most severe problems for freshwater fisheries. Injection and oral vaccines have been developed against BCWD for ayu consisting of formalin-killed cells of *F. psychrophilum*. The disease also occurs in ayu with a weight less than 1.5 g so there is a need for effective immersion vaccines [94]. Previous studies have revealed that development of an immersion-type BCWD vaccine is very difficult. The cell fraction contains important potential candidates for a BCWD vaccine. Additionally, collagenase expressed in *F. psychrophilum* isolates from ayu is considered a promising vaccine candidate [94]. The collagenase enzyme is considered to play an important role in the infection process, but it is difficult to use the native collagenase as a vaccine, since the expression level of *F. psychrophilum* is very low [94]. In this study, the use of recombinant *F. psychrophilum* collagenase as a component in an ayu-BCWD vaccine was investigated [94]. Recombinant *F. psychrophilum* collagenase was expressed in *Brevibacillus chosinensis*. Ayu juveniles were maintained in fresh water and immersed in a collagenase–vaccine solution for 5 min (Experiment I) The culture supernatant of *B. chosinensis* containing mature collagenase was used as a vaccine solution. The experiment was repeated once (Experiment II). The vaccinated fish were challenged by soaking in *F. psychrophilum* culture (4.2 × 10^7^ CFU mL^−1^) for 3 h, and returned to the tanks and reared for another 14 days. In experiment I, the mortality of the vaccinated group and the control group was 17.5% and 47.5%, respectively, and the relative percent survival was 63%. In the second experiment the mortality was 12.5% and 20% in the vaccinated and control group, respectively, with a relative percent survival of 38%.

## 7. Barramundi

### Vibriosis

Culture of barramundi, *Lates calcarifer* (Bloch), is a rapidly growing enterprise in tropical Australia. Systemic vibriosis, caused by the bacterium *Vibrio harveyi,* is a persistent problem in barramundi culture in Australia. In tropical Australia, vibriosis is usually an acute disease affecting fingerling barramundi reared in sea cages, with heavy infections, resulting in fatalities within 24–48 h. This study compared immune responses in barramundi to an experimental *V. harveyi* bacterin administered by various routes: intraperitoneal injection, immersion and anal intubation [75]. Fish were held in sea water (3.5%) at 26.5 °C. *V. harveyi* bacterin was made from cultures of the bacterium, inactivated by formalin to a final concentration of 0.5%. For immersion vaccination, the fish were immersed in a bacterin diluted with sea water (2 × 10^7^ cells mL^−1^) for 60 sec. ELISA was used to quantify specific serum antibody in barramundi after the immunization. The results show that barramundi respond systemically, in terms of antibody, to whole-killed bacterial cell antigens from *V. harveyi* when administered by intraperitoneal injection, immersion or anal intubation [75]. Analyses of antibody activity within groups 21 days after primary immunization showed that the bacterin immunized groups were significantly higher than saline-treated controls, except for the anal-intubated group. Similar patterns were seen at 31 and 42 days post primary immunization, with the exceptions being the immersion immunized group at 31 days and the anal-intubated group at 42 days. Bacteriostatic activity of barramundi serum against *V. harveyi* was observed in all bacterin immunized groups, as well as in the sera of some individuals in control groups.

## 8. Channel/Hybrid/Striped/Vietnamese Catfish

The production of channel catfish (*Ictalurus punctatus*) is >400,000 metric tons annually; half of this production is in the US. Striped catfish (*Pangasius hypophthalmus*) are mainly produced in East Asian countries (Vietnam, Thailand, Cambodia etc.), with an annual production above 500,000 metric tons. As such, this species is of high importance in these countries.

### 8.1. Edwardsiellosis

*Edwardsiella ictaluri* septicemia occurs worldwide and causes high mortality and considerable economic loss to the catfish industry, especially in Vietnam and the USA. To control *Edwardsiella* septicemia, farmers use antibiotics and various vaccination methods. Vaccination with inactivated vaccines has variable efficacy. One study used this approach to control *Edwardsiella* septicemia of Tra catfish (*Pangasanodon hypophthalmus*) in Vietnam via mucosal surfaces. Briefly, catfish weighing 5–6 g and 8–10 g were vaccinated using injection, immersion and oral administration, or combinations of these delivery methods. Catfish were given a primary immersion vaccination (Group A) on day 1 of the experiment (immersion-prime) or a primary vaccination by oral delivery only (Group E, oral-prime) through days 8–21 of the experiment. Group C was given a combination of immersion followed by an oral boost, and Group F was given a second oral boost through experimental days 101–107, 80 days after the first boost. The immersion vaccine consisted of a sterile, water-based, killed bacterial suspension of 5.0 × 10^9^
*E. ictaluri* per ml. The oral vaccine was made by formulating the killed bacteria at a concentration of 3.85 × 10^8^ bacteria per ml in an oil-emulsion, followed by top-dressing on feed pellets [101]. Immersion vaccination was performed by dip vaccination for 1 min in a suspension of 5.56 × 10^8^ bacteria per ml final concentration. The first immersion challenge was performed on day 48 with 4.3–7.6 × 10^6^ bacteria mL^−1^ for 1 h. The second immersion challenge was performed on day 121 with 8.1 × 10^6^ bacteria mL^−1^. After challenge, the fish were observed for 14 days and mortality recorded. The cumulative mortality in the non-vaccinated controls was 87% by day 48. In Group A (immersion-prime) the average cumulative mortality was 65%, while in Group E (oral-prime) the average mortality was 74%, and in Group C (immersion prime/oral boost-1) the cumulative mortality was 42%, giving RPS values of 25, 15 and 52, respectively. Oral vaccinated fish displayed 15% relative survival compared to controls. At experimental day 121, the cumulative mortality in the controls was 90%. In Group A (immersion-prime), cumulative mortality was 80% (RPS 11), while Group E (oral-prime) cumulative mortality showed a mortality of 82% (RPS 9). In Group C (immersion prime/oral boost-1), mortality was 64% (RPS 29), while, in Group F (immersion prime/oral boost-2), cumulative mortality was 48% (RPS 47). The RPS was 9% in the oral vaccine group. In this report, an additional immunization protocol was followed (no.3), where fish were injection vaccinated ± oral boost. Single immunized fish showed a RPS of 3.6%, while orally boosted fish showed a RPS of 11.5%. As a conclusion, combined immersion/oral boost vaccination might be an attractive alternative to protect catfish against lethal challenge with *E. ictaluri* [101]. Contrary to what might be expected, injection vaccination with or without oral boosting induced negligible protection. 

*E. ictaluri* is the causative agent of enteric septicemic of catfish (ESC). Triet et al. (2019) [21] used two *E. ictaluri* wzzE mutants (WzM-L3, deficient in a 1038 bp-entire wzzE gene and WzM-S3, a 245 bp-partial deletion of wzzE). *Pangasius* fingerlings of 5–10 g were immunized in 1.5 × 10^7^ CFU mL^−1^ ofWzM-S3 and 9.7 × 10^6^ CFU mL^−1^ of WzM-L3 for 30 min, and challenged with the wild type *E. ictaluri* strain EIAG of 6.1 × 10^5^ CFU mL^−1^ for 30 min 21 days post-vaccination. Both WzM-S3 and WzM-L3 had a remarkably high protection against ESC, with an RPS of 89.29 and 90, respectively.

Vaccination via immersion allows culturists to inexpensively vaccinate large numbers of small fish, while reducing the stress associated with handling. To minimize costs and maximize vaccine effectiveness without inducing immunological tolerance or suppression [25], finding the appropriate age to vaccinate individual fish species is essential to the aquaculture industry [11]. Various outer membrane proteins of *E. ictaluri* have been found to be immunogenic to channel catfish (*I. punctatus*). Fluorescent microspheres covalently conjugated to a crude extract of *E. ictalurid,* and outer membrane proteins were used to evaluate immunogenic differences—when and where particulate antigens occurred in the mucosa of developing channel catfish [11]. Two forms of carboxylate-modified microspheres (FMS) were used as particulate antigens in a series of immersion exposures (1.0 µm polystyrene). The first form was a carboxylated FMS (blue). The second form consisted of FMS (green) covalently conjugated to a crude extract of *E. ictaluri* outer membrane protein (OMP). Immersions were conducted for 24 h, using an equal mixture of OMP-conjugated FMS (green) and carboxylated FMS (blue) at a concentration of 1 × 10^7^ microspheres mL^−1^. The results showed that both FMS types were observed in the same types of phagocytes—often in the same cells, trafficked to the same locations, and cleared at the same rates [11]. The majority of the FMS uptake occurred in the tissues of the external epithelium and increased with age. Primary sites were head, torso, fins, nares, and, to a lesser extent, the gills.

### 8.2. Columnaris Disease

*Flavobacterium columnare* is a common ubiquitous aquatic bacterium that infects most species of freshwater fish. In the cultured channel catfish industry, it is responsible for significant economic loss [92]. A modified live *F. columnare* vaccine was developed by repeated passage of a virulent strain on increasing concentrations of rifampicin that resulted in attenuation [92,120]. Immersion vaccination of channel catfish (*I. punctatus*) fry between 10 to 48 days old (48 days post-hatch (dph)) resulted in an RPS of 57%–94%. Three trials were performed and, in trial one, channel catfish fry (48 dph) were immersion vaccinated at 28 C at 1 × 10^6^ CFU mL^−1^ or 5 × 10^6^ CFU mL^−1^ for 2 min, followed by 13 min in the vaccine bath diluted 2-fold with water. At 57 days post-vaccination, fish were cohabitated and challenged with three dead fish infected with *F. columnare*. The dead fish were removed after 24 h. The observed cumulative mortality in the sham vaccinated fish after 21 days post-challenge of trial one was 34.7%. Relative percent survival was 96% and 87% in fish vaccinated with 1 × 10^6^ or 5 × 10^6^ CFU mL^−1^, respectively. In the second trial, 7 dph fry were immersed in the vaccine bath for 2 min at the following doses: 4 × 10^5^, 4 × 10^6^, 1 × 10^7^ and 1 × 10^8^ CFU mL^−1^. Following the 2 min exposure, the vaccine bath was diluted 2-fold and the fish were held for 13 min, giving a total immersion time of 15 min. The mortality of the sham vaccinated challenged group was 29.3%. The RPS was 72% in the group immunized with 4 × 10^5^ mL^−1^ of vaccine. The RPS of fish groups vaccinated with the three higher doses ranged from 83% to 87%. In the third trial, 15 dph fry were immersion vaccinated with 1 × 10^7^ CFU mL^−1^ vaccine for 2 min, following dilution to half concentration for an additional 13 min. Total immersion time was 15 min. The mortality of the sham vaccinated fry was 30.7%. Relative percent survival was 57%, against the genome variant II *F. columnare* in the trial [92].

### 8.3. Motile Aeromonas Septicemia (MAS)

Outbreaks of Motile Aeromonas Septicemia (MAS) have cost US catfish aquaculture 60–70 million dollars. This highly virulent *Aeromonas hydrophila* (vAH) pathotype emerged in 2009. Control of vAH is problematic because mortality events on farms are often acute and the mortality is typically seen in larger and highly valuable market-sized fish [99]. Vaccination against the motile *A. hydrophila* strains using formalin-fixed preparations has been practiced, and, at least in laboratory trials, has been shown to be effective in protecting fish against disease. In this study, evaluation of the effectiveness of a simple virulent *A. hydrophila* bacterin, delivered via immersion to hybrid catfish (*Ictalurus punctatus* x *Ictalurus furcatus*), was chosen [99]. In the US, hybrid catfish are being raised more often, due to superior performance traits including faster growth and increased disease resistance. Reports suggest the hybrid catfish is more resistant to virulent *A. hydrophila* than the channel catfish [99,121]. Hybrid catfish with an average weight of 3.79 g were immersed in vaccine for 1 h at 32.8 °C. The vaccine dose was 1.67 × 10^7^ CFU mL^−1^. Challenges were conducted at three, five- and seven weeks post-vaccination. Hybrid catfish immersion, vaccinated with formalin-killed bacterin, were 90% protected 3 weeks post-vaccination. Similar results were seen 5 weeks post- vaccination, with survival of 83–88%. The 7-week survival was 93.3%, compared to the mock vaccinated fish survival of 61.7%, against a more recent, industry-relevant isolate of virulent *A. hydrophila* strain (ALG-15-097), to determine if cross-protection was observed [99].

## 9. European Eel

### Vibriosis

*Vibrio. vulnificus* is a heterogenous species that comprises at least four serovars, pathogenic for eels [76,122]. Vulnivaccine, a vaccine against *V. vulnificus*, has been shown to protect eels against vibriosis after vaccination by triple prolonged immersion at glass eel stage [76]. The main objective of a study by Esteve-Gassent et al. (2004) [123], was to evaluate the efficacy of Vulnivaccine (licensed by the University of Valencia, Spain [123]) as an oral booster after the immersion vaccination of glass eels. Elvers (*Anguilla anguilla* L.)—average body weight 4.50 g—had been subjected to thrice one-hour immersion vaccination 6 months earlier. The infective dose was 10^7^ CFU mL^−1^ when bath challenged. RPS values were 82.5, 80.0, 73.4 and 75, at 11, 15, 30 and 60 days after oral booster, respectively.

## 10. Japanese/Olive Flounder

### 10.1. Edwardsiellosis

*Edwardsiella tarda* is one of the important bacterial pathogens for aquacultured fish, especially for the cultured flounder (*P. olivaceus*) industry in Asia. Flounders of 15–17 cm length were immersion vaccinated with an inactivated *E. tarda* bacterin at 10^6^, 10^7^, 10^8^ and 10^9^ CFU mL^−1^, for 30, 60 and 90 min, respectively [14]. At the sixth week post-vaccination, the fish were challenged with live *E. tarda*, and the relative percent survival were 70, 78, 74 and 65 for vaccination groups at 10^9^-30 min, 10^8^-60 min, 10^8^-90 min and 10^7^-90 min. Formalin-killed *E. tarda* bacterin was prepared in four concentrations: 10^9^, 10^8^, 10^7^ and 10^6^ CFU mL^−1^. Japanese flounder were immersed in the various vaccine suspensions for 30, 60 and 90 min. Quantitative real-time PCR was employed to examine the bacterial uptake in the gill, skin, spleen and kidney [14]. The results showed that antigen uptake in gills and skin were significantly higher than in spleen and kidney. Significantly higher amounts were detected in 10^9^-30, 10^9^-60, 10^8^-60, 10^8^-90 and 10^8^-90 groups. Also, the expression of immune-related genes was upregulated in fish that received 10^7^-90, 10^8^ and 10^9^ CFU mL^−1^.

Flounder were immersed in three hyperosmotic solutions of 40, 50 and 60 ^o^/_oo_ salinities, then transferred to seawater of 30 ^o^/_oo_ salinity containing formalin-inactivated *E. tarda* for 30 min. Antigen uptake was determined by quantitative PCR, and the results showed a significantly higher uptake in flounders immersed in solutions of 50 and 60 ^o^/_oo,_ compared to the control group immersed directly in the vaccine [16]. A rapid and significant increase was detected in the gill, skin and intestine, compared with the spleen, kidney and liver. The expression of immune-related genes (MHCIα, MHCIIα CD-4-1 and CD8α were increased in the flounders exposed to 50 ^o^/_oo_, salinity compared to the control group.

### 10.2. Vibriosis

Flounders weighing 35 g were kept at 20 °C in running seawater. For immunization, fish were immersed in inactivated *V. anguillarum* bacterin at a final concentration of 1 × 10^8^ CFU mL^−1^ for 30 min. Polymeric immunoglobulins (Igs) and polymeric immunoglobulin receptor (pIgR) play crucial roles in teleost mucosal surface defenses. In this study, pIgR and IgM responses were analyzed [78]. Real-time PCR showed that pIgR and IgM mRNA expression were upregulated. The pIgR responded earlier than IgM and the pIgR mRNA levels were higher in the spleen, gills, skin and hindgut. Immunohistochemical studies revealed that pIgR and IgM were localized in the epithelium of the skin, gills and hindgut and the biliary epithelium of the liver [78]. 

### 10.3. Viral Hemorrhagic Septicemia (VHS)

Vaccination by immersion is suitable for mass vaccination of small fish. To date, very few viral vaccines have been developed for immersion application because of low efficacy. VHS is one of the most important viral diseases in the olive flounder aquaculture industry in Korea/Japan, and in salmonid fish in European aquaculture. The efficacy of an immersion vaccine against VHS, containing Montanide IMS1312 VG adjuvant (SEPPIC, France), in olive flounder [123] has been evaluated following immersion administration. Fish of 14.1 cm and 25.5 g were immersion vaccinated with a heat-inactivated strain of VHS containing Montanide IMS1312 VG for 5 min at 20 °C in 5L buckets. This adjuvanted vaccine enhanced gene expression of immune-associated genes, interleukin-(IL)-1ß, IL-6, IL-8, and Toll-like receptor (TLR)-3. On weeks 4 and 8 after vaccination, fish were challenged intraperitoneally with VHSV (10^6^ TCID_50_ per fish). Cumulative mortality reached 90% on day 14 after challenge of control fish. Fish immersion vaccinated in the absence of adjuvant showed 70% cumulative mortality, and fish vaccinated in the presence of 10 g and 50 g Montanide mixed with 10^7^ TCID_50_ showed a considerable delay in the onset of mortality, and cumulative mortalities were 10% and 25%, respectively. Eight weeks after vaccination, fish vaccinated in the presence of 10 g and 50 g premixed with Montanide IMS 1312 and inactivated VHS gave RPS values of 67% and 47%, respectively, which indicated that the protective effect was short-lived [7].

In another study, efficacy of two recombinant attenuated VHSV strains passaged in two commercially available cell lines, EPC and RTG-2 were evaluated. A4G-G5A showed an attenuated growth profile in both the EPC and RTG-2 cell lines, whereas the growth profile of ∆NV was comparable to the wild-type (WT) strains in RTG-2 cells in contrast to EPC cells. Juvenile olive flounders (average 13 cm, 23 g) were immersion vaccinated with 10^2.5^, 10^3.5^, 10^4.5^ and 10^5.5^ TCID_50_ mL^−1^ of A4G-G5A and then boost immunized one week later with the same regime as for the first immunization, giving 5%–13.3% cumulative mortality. Immunization was followed by an intramuscular challenge 35 days after the first immunization using VHSV-WT at 10^5^ TCID_50_ per fish. The relative percent survival (RPS) in immunized groups ranged from 81.6% to 100% which demonstrated a high level of protection [108].

## 11. Grouper/Sevenband Grouper

### 11.1. Viral Nervous Necrosis (VNN)

Viral nervous necrosis (VNN) affects more than 34 species of aquacultured fish, including grouper. The causative agent is a non-enveloped nodavirus. Grouper is an economically important species in the aquaculture industry of Taiwan. In this study, bath immunization of grouper larvae (*Epinephelus coioides*) against betanodavirus was performed [112]. Larvae of 0.2 g, with a total length of 2.4 cm, were bath immunized with formalin and binary ethylenimine inactivated betanodavirus at a dose of 10^7^ TCID_50_ mL^−1^. In another set of experiments, the effect of various virus doses was tested at 10^5^, 10^6^ and 10^7^ TCID_50_ mL^−1^. Also, the effect of immersion time was investigated, using 20, 60- and 120 min bathing time. Bath challenge tests were done 30 days post-immunization with a dose of 1.6 × 10^6^ TCID_50_ mL^−1^. The cumulated mortality was recorded one month post-challenge. The cumulative mortality of control fish was 87%, 53% for 0.1% formalin-inactivated and 50% for 0.2% formalin-inactivated and 19% for binary ethylenimine-inactivated vaccine groups. The RPS values of the group immunized with ethylenimine-inactivated VNN was 79, compared with 39 for formalin-inactivated vaccine. Larvae immunized with final concentrations of 10^5^, 10^6^ and 10^7^ TCID_50_ mL^−1^ ethylenimine-inactivated VNN, gave RPS values of 7.5, 87.5 and 95, respectively. Fish immunized with 10^7^ TCID_50_ mL^−1^ for 20, 60 and 120 min resulted in RPS of 75, 95 and 88. In another set of experiments, grouper larvae bath-immunized with binary ethylenimine-inactivated vaccine with a final concentration of 10^7^ TCID_50_ mL^−1^ were challenged with VNN with a titer of 1.6 × 10^6^ TCID_50_ mL^−1^ on the 15th, 30th and 90th day post-vaccination. This regime resulted in RPS values of 30, 87 and 82, respectively [112]. 

It was reported 10 years ago that polyinosinic polycytidylic acid, poly (I:C), stimulation of fish conferred non-specific/innate protection against nodavirus infection [124]. Poly (I:C) is a synthetic, double-stranded DNA known to induce innate defense mechanisms, especially antiviral ones. It mimics a viral infection and has, therefore, been used to induce IFN type I in many fish species [125]. 

The process of poly (I:C)-induced protection involved an immunization with live pathogenic virus, followed by administration of poly (I:C), which induces a transient, non-specific/innate antiviral state in fish. As a result, the fish survived the initial immunization with live pathogenic virus, which would otherwise be lethal. This study reported on determination of the exact dosage of red-spotted grouper nervous necrosis virus (RGNNV) required for poly (I:C) adjuvanted immunization of sevenband grouper (*Epinephelus septemfasciatus*) [113]. To obtain more than 90% relative percent survival, 10^5.3^ TCID_50_ per fish or greater of RGNNV was required for immersion vaccination. The degree of RGNNV infection must be similar to a fatal dose in order to become immune. Antibodies against RGNNV were not detected in sera from fish immunized by immersion.

This study reports on the potentiality of live nervous necrosis virus (NNV) vaccine for sevenband grouper at a low rearing temperature (17 °C) compared to the optimum temperature of 26 °C. Fish mortality was reduced by decreasing the fish-rearing temperature, and no mortality was observed in fish reared at 17 °C, regardless of the infection method [114]. During the increment of temperature from 17 °C to the optimum temperature of VNN onset (26 °C), increased mortalities were observed in the survivors from the first NNV infection. Little or no mortality was observed in the second NNV infection. This demonstrates that the survivors of the first NNV-infection mounted a specific protective response against NNV (survival rate of 93.3%).

### 11.2. Iridovirus

Iridovirus is one of the most devastating viral pathogens in grouper (*Epinephelus* spp.). This article reports the use of a new adjuvant in an immersion GIV subunit vaccine (Major capsid protein (MCP). The vaccine enhanced the survival of infected fish in a dose-dependent manner. Two weeks after immersion vaccination, MCP antibodies were detected. Boosters at 1 or 2 weeks after the initial vaccination enhanced the yield of specific antibodies and the protection against GIV both 3 and 4 weeks after the initial vaccination [117].

## 12. Guppy, Gourami

### Viral Nervous Necrosis/Viral Encephalopathy and Retinopathy Virus

VER virus, a betanodavirus, causes mass mortalities in warm-water environments. The nervous necrosis virus causes disease in fish at a very early age, when the injection immunization is impractical or impossible. In this study, the immune response of freshwater fish, guppy, *Poicelia reticulate*, and gourami, *Trichogaster tricopterus* against the recombinant coat protein of *Epinephelus tauvina* nervous necrosis virus (ETNNV), and formalin-inactivated ETNNV by immersion immunization, was focused on. In vitro neutralization using the whole-body extract of immersion immunized guppy showed the importance of recombinant coat protein as a candidate vaccine against viral nervous necrosis and its suitability for immersion delivery [115].

## 13. Koi/Common/Grass carp

### Koi Herpes Virus

Koi herpes virus infects and causes mass mortality in koi and carp. This study reports on the efficacy of koi herpes virus DNA vaccine by immersion administration on *Cyprinus carpio* [116]. Thirty-day old common carp juveniles of mean body weight of 0.19 g were immunized for 30 min in water (21 °C) containing 1.3 × 10^8^ CFU mL^−1^ of heat-killed *Escherichia coli* carrying DNA vaccine encoding glycoprotein-25. Two different densities of fish were immersion vaccinated: 800 fish L^−1^ (V^8^) and 1200 fish L^−1^ (V^12^). The challenge test was performed 30 days post-vaccination by intramuscular injection of koi herpes virus filtrate. The cumulative mortality of the (V^12^) amounted to 82.2%, while the (V^8^) fish mortality was 31.2%, resulting in a relative percent survival of 9.89 and 63.3, respectively [116]. 

## 14. Sea Bass

Sea bass (*Dicentrarchus labrax*) is an important aquacultured species in the Mediterranean with an annual production of more than 191,000 metric tons. 

### 14.1. Vibriosis and Pasteurellosis

Vibriosis is considered to be the main disease affecting sea bass [126]. To cope with vibriosis, several vaccine approaches have been taken. In one study, sea bass fry (1.5 g, 65 dph) were immunized using a commercial bivalent vaccine (Aquavac Vibrio VAB, Schering Plough) containing *Listonella* biotypes O1 and O2, 10^9^ cells mL^−1^ [81]. In the experimental group, 1: 95 dph, 1.5 g fry were immersion primed for 1 min with the vaccine diluted 1:10 in sea water. Experimental group 2: 95 dph, 1.5 g fry were primed as above and immersion boosted for 1 min at 16 g, 165 dph. Two challenge experiments were performed with *L. anguillarum* serotype O1. In the first experiment, 298 dph fish were i.p. challenged with 200 µL of a 2.4 × 10^6^ CFU mL^−1^. In the second experiment, each fish (309 dph) was i.p. injected with 200 µL of a 6.2 × 10^6^ CFU mL^−1^. The authors do not reveal the RPS of the single immersion vaccinated fish, but the immersion boosted fish with the lowest infective dose was 78, and 74 for the highest infective dose [81].

Sea bass juveniles (mean weight 3.3 g) were reared in sea water at 18 °C. The vaccine contained dead *V. anguillarum*. A killed *V. anguillarum* bacterin was tested by immersion vaccination for 60 s with a booster vaccination of a subset of fish at 60 days. Thirty days after booster vaccination, fish were challenged with virulent *V. anguillarum* bacteria (3.0 × 10^6^ CFU fish^−1^). Twenty days after the challenge, the mortality was 0% among the booster vaccinated, 10% among the fish vaccinated once and 50% of unvaccinated control fish [77]. 

In another study, the relative degree and duration of protection against *V. anguillarum* and *Patsteurella piscicida* in sea bass vaccinated once by immersion at 1 g and revaccinated at 5 g by immersion [80] was determined. The vaccines consisted of formalin-inactivated *V. anguillarum* serotype O1 and *P. piscicida*. The fish were immunized for 30 sec and challenged. Fish were challenged with both pathogens separately at 7- and 26 weeks post revaccination by intraperitoneal injection with *V. anguillarum* and immersion challenge for *P. piscicida*. Sea bass vaccinated by immersion at 1 g and revaccinated by immersion at 5 g demonstrated a significant level of protection against *V. anguillarum,* lasting for 26 weeks post revaccination. Sea bass revaccinated by immersion were significantly protected against *P. piscicida* 7 weeks post revaccination. One gram fish not revaccinated did not demonstrate protection against either pathogen [80]. 

### 14.2. Viral Nervous Necrosis (VNN)/Viral Encephalopathy and Retinopathy Virus (VER)

In order to exploit whether an immersion vaccine would confer protection against a problematic viral disease caused by betanodavirus, fish (average weight of 2.1 g) were immersion vaccinated in a 30 L tank containing formalin inactivated VER 10^6^ TCID mL^−1^ for 2 min. The fish were experimentally challenged with VER. This study showed that the immersion treatment induced low levels of specific IgM production and gave no protection compared to control fish [127]. While immersion vaccines against VER have not been proved to exhibit high efficacy, they have induced gene expression of relevant adaptive and antiviral genes [128].

## 15. Hybrid Striped Bass

The production of hybrid striped bass (HSB; *Morone chrysops* × *Morone saxatilis*) reached 5 kilotons in 2016, with the US being the main producer. 

### Streptococcosis

Juvenile hybrid striped bass were vaccinated through bath immersion and held for 800 degree-days prior to challenge with a lethal dose of the virulent wild-type (WT) *S. iniae* parent strain. *S. iniae* WT strain was isolated from the brain of diseased HSB. Allelic replacement mutants, ∆simA [129] and ∆cpsD [130], and the ∆pgmA transposon mutant [131] were generated from the strain K288 background. The fish were immersion vaccinated for 90 min in 30 × 10^7^ CFU mL^−1^. After 800 degree-days, the fish were challenged intraperitoneally with 1 × 10^6^ CFU of the WT K288 strain and monitored for survival for 2 weeks post-challenge [104]. This study demonstrated the efficacy of live attenuated vaccines for prevention of *S. iniae* infection. The ∆cpsD immersion vaccinated group displayed an RPS value of 51%, whereas the RPS of fish administered with the bacterin had an RPS of 19%. The ∆simA mutant was the only vaccine to achieve 100% RPS.

## 16. Gilthead Sea Bream

Sea bream (*Sparus aurata*) is an important fish species for the aquaculture industry, especially in the Mediterranean countries, with an annual world-wide production of above 185,000 metric tons.

### Pasteurellosis 

Pasteurellosis is one of the main bacterial diseases causing severe economic loss of sea bream. Formalin and heat-treated toxoid-enriched extracellular products, together with the formalin inactivated *P. piscicida* administered as an immersion vaccine, conferred protection against experimental challenge with *P. piscicida*, with an RPS of approximately 40%. Booster vaccination did not induce any further survival or any cross-protection against *V. anguillarum* [132]. Furthermore, a separate study investigated the effect of the inactivation process of vaccines on the expression levels of proinflammatory and Mx genes after immersion vaccination of sea bream against *P. damselae* subsp. *piscicida*. The following inactivated vaccines were prepared: a formalin killed vaccine, a heat-shock treated vaccine (80 °C for 10 min) and a UV-light treated vaccine. Sea bream juveniles with an average weight of 5 g were immunized by immersion in a 10-fold dilution of each bacterin (10^8^ CFU mL^−1^) for 60 sec. The results showed that the heat-inactivated vaccine stimulated an up-regulation of IL-1ß, type II IL-1R, Cox-2 and TNF-α genes, whereas the inactivated UV-light stimulated the expression of the Mx gene [98]. This study did not determine vaccine efficacy.

## 17. Nile/Red Hybrid Tilapia

Aquaculture of tilapia is mostly restricted to sub- and tropical areas, where China is the biggest producer with respect to world production, with over 4 million metric tons in 2015 [133]. Tilapia production consists of *Oreochromis niloticus, O. mosssambicus, O. aureus, O. urolepis hornorum*, Tilapia rendalli and zilli. In aquaculture, the tilapias may suffer from several bacterial disease caused by, e.g., *F. columnare*, *E. tarda*, *Aeromonas* spp., *Vibrio* spp., *Francisella* spp., *Streptococcus agalactiae* and *S. iniae* [134]. In addition, several different viruses may infect tilapia and cause diseases [133,135]

### 17.1. Streptococcosis, Lactococcosis, Enterococcosis

Immersion vaccines account for a small proportion of vaccines used against *S. agalactiae* compared to injectable vaccines in tilapia [106]. Live attenuated vaccines are more protective than inactivated vaccines. One approach for generating avirulent strains is serial passage of pathogenic strains on growth media. Li et al (2015) showed that a serially passaged pathogenic strain of *S. agalactiae* lost its virulence. Immersion vaccination with live *S. agalactiae* attenuated vaccine (YM001) gave an RPS of 67.22%. Tilapia immersed vaccinated in formalin-inactivated bacteria resulted in an RPS of 34% after challenge [106]. 

The effectiveness of a formalin-inactivated *S. agalactiae* vaccine in tilapia (*O. niloticus*) after bath immersion was conducted at a temperature of 32 °C, and with a mean fish weight of 5 and 30 g. The CFU of *S. agalactiae* in the final vaccine preparation was estimated to 4 × 10^9^. Bath immersion vaccination of 5 g and 30 g tilapia, followed by intraperitoneal *S. agalactiae* challenge at 3.6 × 10^5^ and 1.7 × 10^6^ CFU per fish, respectively produced an RPS of 34% and 35 % [105].

Streptococcosis, lactococcosis and enterococcosis are the most frequent diseases affecting tilapia production in Egypt. A vaccine against streptococcosis, lactococcosis and enterococcosis was formulated as a polyvalent inactivated vaccine, containing *S. agalactiae*, *S. iniae*, *Lactococcus garviae* and *Enterococcus faecalis* along with a nano-particulate adjuvant. Virulent strains of *S. agalactiae*, *S. iniae*, *L. garviae* and *E. faecalis* were isolated from Nile tilapia, cultured in brain heart infusion broth at 30 °C, and inactivated with formalin. To formulate the polyvalent vaccine, more than 10^9^ CFU mL^−1^ of each bacterial species was mixed with the Montanide IMS 1312 VG adjuvant. One-month old tilapia fry were immersed in the vaccine for 2 min at 25 °C. To evaluate the resistance of vaccinated fish, experimental infection with virulent strains of *S. agalactiae*, *S. iniae*, *L. garviae* and *E. faecalis* was performed in four different groups; each received a single homologous bacterial species by intraperitoneal injection of 0.2 mL × 6 × 10^8^ CFU mL^−1^. Each group was further divided into three subgroups for challenge after 1, 2- and 3 months post-vaccination [107]. Challenge results obtained monthly after immersion vaccination confirmed that the vaccine provided protection against the included pathogens for up to three months post-vaccination. 

### 17.2. Columnaris

A mucoadhesive vaccine delivery system to enhance the efficacy of direct immersion vaccination against *F. columnare*, the causative agent of columnaris disease was tested in red tilapia [17]. A formalin-killed negatively charged bacterial cell suspension was used to prepare a mucoadhesive vaccine by electrostatic coating with positively charged chitosan. The main aim of this study was to investigate the application of chitosan to facilitate efficient delivery of inactivated vaccines to fish mucosal surfaces. Red tilapia fingerlings (*Oreochromis* sp.) with an average weight of 10 g were immersed in 10^7^ CFU mL^−1^ of vaccine preparation for 30 min. At 30 and 60 days after vaccination, the fish were immersion challenged with 1 × 10^6^ CFU mL^−1^ of a virulent strain of *F. columnare* for 1 h. Cumulative mortality was recorded for 10 days after immersion challenge. At 30 days post-vaccination with naked and chitosan vaccines, RPS was 4% and 81%, respectively [17], and mortality of non-vaccinated and naked vaccine groups was 90% and 87%, versus 17% mortality in the chitosan vaccine group. Sixty days after vaccination, the RPS was reduced to 58.

### 17.3. Francisellosis

*Francisella noatunensis* subspecies *orientalis* (Fno) (syn. *F. asiatica*) is a small pleomorphic Gram-negative coccobacillus recently described as the causative agent of piscine francisellosis in an increasing number of cultured fish species. In tilapia (*Oreochromis* sp.), francisellosis can present as an acute disease with few clinical signs and high mortality rates or as a sub-acute to chronic infection with non-specific clinical signs. The bacterium is highly infective to tilapia fingerlings [10]. Hybrid tilapia (*O. niloticus* x *O. mosssambicus*) fry of 5 g and fingerlings of 10 g were maintained in fresh water at either 25 or 30 °C. Fish were bath vaccinated in a formalin-killed suspension of 7 × 10^5^ CFU mL^−1^ bacteria for 3 h. At four weeks post-vaccination, fish were challenged at 25 °C with 4 × 10^3^ CFU mL^−1^ wild type Fno for 1 h. Because francisellosis does not occur naturally in tilapia at 30 °C, it was necessary to reduce water temperatures prior to challenge. Mortality was monitored twice daily for a period of 4 weeks. The relative percent survival of 5 g hybrid tilapia fry vaccinated at 25 and 30 °C was −4% and 30%, respectively, compared to 35% and 38% in the 10 g fingerlings [10].

## 18. Mandarin Fish 

### Infectious Spleen and Kidney Necrosis Virus

Mandarin fish (*Siniperca chuatsi*) are strict carnivores and their production in China is greater than 300,000 metric tons. Infectious spleen and kidney necrosis virus (ISKNV) cause a high mortality disease, which leads to significant economic loss of mandarin fish and Chinese perch. There is no effective vaccine against this fatal disease at present. The effect of an immersion subunit vaccine system (SWCNTs- major capsid protein (MCP)) encoding *MCP* gene of ISKNV based on single-walled carbon nanotubes (SWCNTs) was determined [118]. Juvenile Mandarin fish weighing 6 g were vaccinated via bath administration for 10 h. On the 28th day post-vaccination, fish were challenged by intraperitoneal injection with 50 µL of 3.98 × 10^6^ TCID_50_ mL^−1^ live ISKNV. The results showed increased protection over the naked subunit vaccine by 28.1% 

## 19. Common Carp/Grass Carp

Annual world production of common carp (*C. carpio*) is estimated to be more than 4.5 million metric tons (2016), with 10% produced in Europe (FAO: http://www.fao.org/fishery/culturedspecies/Cyprinus_carpio/en). The production contributes to approximately 14% of the total global freshwater aquaculture. Since production is widely distributed, common carp may suffer from diverse diseases. 

Immunoprophylactic measures were already tried out in 1985 by Lamers et al. [136], where carp were immersed in water containing *A. hydrophila* bacterin. This study showed that carp developed an antibody response, and speculated that a particular mucosal response was present. 

### 19.1. Koi Herpes Virus

Common carp (0.19 g) was immersed in water containing 1.3 × 10^8^ CFU mL^−1^
*E. coli* expressing GP-25 (koi herpes virus glycoprotein 25) for 30 min. Results showed higher survival compared to controls, with an RPS of 63.37, after experimental challenge [116].

### 19.2. Spring Viremia of Carp 

Spring viremia of carp virus (SVCV) has caused mass mortality in cyprinids, especially in young fish. To enhance the efficacy of immersion vaccination, functionalized single-walled carbon nanotubes (SWCNTs) were used as a carrier to manufacture an immersion DNA vaccine system (SWCNTs-pEGFP-M) encoding the vp7 gene of SVCV. Common carp (*C. carpio*) weighing 4 g were exposed to pEGFP-M and SWCNTs-pEGFP-M by immersion for 10 h at 20 °C. At 3 days post-immunization, fish were booster immunized with the same regime as the primary immunization [111]. Before virus challenge, the water temperature was gradually lowered to 15 °C and 21 days post-vaccination the fish were intraperitoneally injected with 50 µL of 6.0 × 10^4^ TCID_50_ mL^−1^ of live SVCV. The results show that SWCNTs-pEGFP-M is a promising vehicle and can enhance the RPS by 23.8%, compared with naked pEGFP-M immunized fish. Apparently, the use of SWCNTs as vehicles for immersion vaccines can induce high and long-term specific antibody levels [111]. There is, however, a debate as to whether SWCNT may cause toxicity in fish effects following immersion, as noted for rainbow trout [137].

### 19.3. Flavobacteriosis

A novel attenuated *Flavobacterium johnsoniae* M170 vaccine was developed from a pathogenic *F. johnsoniae* M168 isolate using a streptomycin-resistant strategy. Grass carp (*Ctenopharyngodon idella*) with a mean weight of 10.1 g were bathed in 1 × 10^7^ CFU mL^−1^ of the attenuated vaccine for 30 min. At 28 and 240-days post-vaccination (dpv) the fish were challenged with virulent *F. johnsoniae* M168 by immersion in 1 × 10^7^ CFU mL^−1^ for 30 min [93]. The results showed that fish vaccinated with the attenuated strain had a RPS of 73.1 and 60 when challenged 28 days post-vaccination (dvp) and 240 dpv, respectively, meaning that protection lasted for up to 8 months.

## 20. Turbot

The annual aquaculture production of turbot (*Scophthalmus maximus/Psetta maxima*), mainly in Europe (Spain and France), was below 65,000 metric tons in 2016. China also produces this species.

### 20.1. Enterococcosis

Toranzo et al. (1995) isolated toxoid-enriched whole cell bacterin for immersion vaccination. The turbots were immersed in water containing 10^8^ inactivated bacteria per ml for 1 min. The fish were challenged with *Enterococcus* sp. (RA-99.1 strain) by intraperitoneal injection. This vaccination regime did not result in increased survival compared with control fish [138]. In line with this finding, the Aquavac Furovac 5 and an autogenous vaccine was not effective to protect turbot by immersion vaccination when challenged with *A. salmonicida* [139].

### 20.2. Edwardsielllosis

An *E. tarda* attenuated vaccine, EIB202 mutant WED, with deletions in the T3SS genes for EseB, EseC, EseD and EseA with the *aro*C gene for the biosynthesis of chorismic acid, as well as the curing of endogenous plasmid pEIB202, was constructed. Compared to wild-type EIB202, which was highly virulent towards turbot, the WED vaccine was highly attenuated and exhibited impaired capacity to survive in a fish tissue [103]. Turbot of approx. 30 g were vaccinated by immersion for 1 h in a cell concentration of 1 × 10^7^ CFU mL^−1^. At 5 weeks, 2, 4, 6 and 12-months post-vaccination, fish were challenged with 1 × 10^3^ CFU g^−1^ (approx. 2xLD_50_) of the wild type *E. tarda* EIB202. Fish displayed an RPS of 35.7–63.3 during the 6-month investigation. 

## 21. Senegalese Sole

Portugal, Spain, France and Italy are the main producers of common sole (*Solea senegalensis*). The world production of common sole (*Solea solea*) and *S. senegalese* reached 148 metric tons in 2016.

Senegalese sole suffers from infection with *P. damselae* spp. *piscicida*. A study undertaken by Nunez-Diaz et al. (2017), where Senegalese sole (52 g) was immersion -vaccinated using a monovalent formalin, treated *P. damselae* together with inactivated extracellular products. The fish were bathed in a 1/10 dilution of the vaccine for 50 min. Ninety days after immunization, the fish were experimentally challenged with homologous pathogen (1.2 × 10^4^ CFU per fish). The result showed that immersion-vaccinated fish gained an RPS of 35% whereas, in comparison, the i.p injected group displayed full protection [140].

## 22. Conclusions

Vaccination by immersion is typically inferior for mass vaccination of small fish and varies depending on the methods of preparation and species of fish. Vaccination modality has to be optimized for each fish species, as there is no generality for the development of effective immersion vaccines. Immersion vaccines should be developed to deliver antigens to target tissues more effectively. Immersion vaccine efficacy may also be increased by using molecular adjuvants, vaccine carriers and physical methods. It appears that the most promising modality to elicit protection is to use live attenuated pathogens, but rigorous risk assessment is required prior to commercialization. Intraperitoneal and intramuscular challenge methods are often required for efficacy testing, but these do not mimic natural infections because the infective agents do not gain entry through the natural portals of entry. Future studies should seek to develop and use more relevant and suitable challenge models that mimic natural infections.

## Figures and Tables

**Table 1 microorganisms-07-00627-t001:** Central literature on immersion vaccination and selected background information on pathogens.

Pathogen	Disease	Fish species	Vaccine	Reference
***Vibrio harveyi***	Vibriosis	Barramundi/*Lates calcarifer*	Inactivated	[75]
***Vibrio vulnificus***	Vibriosis	European eel/*Anguilla anguilla*	Inactivated	[76]
***V. anguillarum***	Vibriosis	Sea bass/*Dicentrarchus labrax*	Inactivated	[77]
***V. anguillarum***	Vibriosis	Flounder/	Inactivated	[78]
***V. anguillarum***	Vibriosis	Atlantic cod/*Gadus morhua*	Inactivated	[73]
***V. anguillarum***	Vibriosis	Rainbow trout/*Oncorhynchus mykiss*	Inactivated	[79]
***V. anguillarum Pasteurella piscicida***	Vibriosis Pasteurellosis	Sea bass/*Dicentrarchus labrax*	Inactivated	[80]
***L. anguillarum***	Vibriosis	Sea bass	Inactivated	[81]
***Yersinia ruckeri***	Yersiniosis	Atlantic salmon/*Salmo salar*	Inactivated	[82]
***Y. ruckeri***	Yersiniosis	Atlantic salmon	Inactivated	[83]
***Y. ruckeri***	Yersiniosis	Rainbow trout/*Oncorhynchus mykiss. mykiss*	Inactivated	[84]
***Y. ruckeri***	Yersiniosis	Rainbow trout	Inactivated	[85]
***Y. ruckeri***	Yersiniosis	Rainbow trout	Inactivated	[86]
***Y. ruckeri***	Yersiniosis	Rainbow trout	Inactivated	[87]
***Y. ruckeri***	Yersiniosis	Rainbow trout	Inactivated	[88]
***Y. ruckeri***	Yersiniosis	Rainbow trout	Inactivated	[89]
***Y. ruckeri***	Yersiniosis	Rainbow trout	Inactivated	[6]
***Y. ruckeri***	Yersiniosis	Rainbow trout	Lipopolysaccharide	[90]
***Y. ruckeri Flavobacterium columnare***	Yersiniosis/Enteric Redmouth Disease/ERM Columnaris	Atlantic salmon/S. salar Coho salmon/*Oncorhynchus kisutch* Rainbow trout/*O. mykiss*	Inactivated	[91]
***F. columnare***	Columnaris	Channel catfish/*Ictalurus punctatus*	Attenuated	[92]
***F. columnare***	Columnaris	*Oreochromis* spp.	Inactivated	[17]
***F. johnsoniae***	Columnaris	*Grass carp/Ctenopharyngodon idella*	Attenuated	[93]
***Flavobacterium psychrophilum***	Bacterial Coldwater Disease (BCWD)	Ayu/*Plecoglossus altivelis*	Inactivated	[94]
***F. psychrophilum***	Bacterial Coldwater Disease (BCWD)	Rainbow trout	Attenuated	[95]
***F. psychrophilum***	Rainbow trout fry syndrome	Rainbow trout	Live attenuated	[96]
***F. psychrophilum***	Rainbow trout fry syndrome	Rainbow trout	Live non-attenuated	[97]
***Photobacterium. damselae***	Pseudotuberculosis	Sea bream/*Sparus aurata*	Inactivated	[98]
***Aeromonas hydrophila***	Motile Aeromonas septicemia (MAS)	Hybrid catfish/*Ictalurus furcatus* x *Ictalurus punctatus*	Inactivated	[99]
***Aeromonas salmonicida***	Furunculosis	Salmonids	Live	[100]
***Edwardsiella ictaluri***	Edwardsiellosis	Channel catfish/*I. punctatus*	Outer membrane proteins	[11]
***E. ictaluri***	Edwardsiellosis	Vietnamese catfish/*Pangasanodon hypophthalmus*	Inactivated	[101]
***Edwardsiella tarda***	Edwardsiellosis	Japanese/Olive flounder/*Paralichthys olivaceus*	Inactivated	[102]
***E. tarda***	Edwardsiellosis	Japanese flounder	Inactivated	[14]
***E. tarda***	Edwardsiellosis	Flounder/*P. olivaceus*	Inactivated	[16]
***E. tarda***	Edwardsiellosis	*Pangasius hypophthalmus*	Live attenuated	[21]
***E. tarda***	Edwardsiellosis	Turbot/*Scophthalmus maximus*	Live attenuated	[103]
***Streptococcus iniae***	Streptococcosis	Hybrid striped bass/*Morone chrysops* x *Morone saxatilis*	Attenuated	[104]
***S. iniae***	Streptococcosis	Rainbow trout/*O. mykiss*	Inactivated	[12]
***Streptococcus agalactiae***	Streptococcosis	Tilapia/*Oreochromis niloticus*	Inactivated	[105]
***S. agalactiae***	Streptococcosis	Tilapia	Attenuated	[106]
***S. agalactiae*** ***S. iniae*** ***Lactococcus garviae*** ***Enterococcus faecalis***	Streptococcosis Lactococcosis Enterococcosis	Tilapia	Inactivated	[107]
***Francisella noatunensis* subsp. *Orientalis*, syn: *Francisella asiatica***	Francisellosis	Hybrid tilapia/*O. niloticus* × *Oreochromis mossambicus*	Inactivated	[10]
**VHSV**	Viral hemorrhagic septicemia	Flounder/*P. olivaceus*	Inactivated	[7]
**VHSV**	Viral hemorrhagic septicemia	Flounder	Recombinant attenuated	[108]
**VHSV**	Viral hemorrhagic septicemia	Rainbow trout	DNA vaccine	[109]
**VHSV**	Viral hemorrhagic septicemia	Rainbow trout	DNA vaccine	[110]
**SVCV**	Spring viremia of carp	Common carp/*Cyrpinus carpio*	DNA	[111]
**VNNV**	Viral nervous necrosis (VNN)	Orange-spotted grouper/*Epinephelus coioides*	Inactivated	[112]
**Red-spotted grouper NNV (RGNNV)**	Viral nervous necrosis	Sevenband grouper/*Epinephelus septemfasciatus*	Live	[113]
**NNV**	Viral nervous necrosis	Sevenband grouper	Live	[114]
**Epinephelus tauvina nervous necrosis virus (ETNNV)**	Viral nervous necrosis	Guppy/*Poicelia reticulate* Gourami/*Trichogaster tricopterus*	Inactivated/recombinant coat protein	[115]
**Koi herpes virus**		Common carp/*C. carpio*	DNA vaccine	[116]
**Iridovirus**		Grouper/*Epinephelus* sp.	Subunit	[117]
**ISKNV**	Infectious spleen and kidney necrosis	Mandarin fish/*Siniperca chuatsi*	Subunit	[118]

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
