# Peer review of "(untitled)"

_microorganisms, 2019, doi:10.3390/microorganisms7120627_

Round 1

Reviewer 1 Report

Effective immersion vaccines, together with other types of mucosal vaccines (oral) are urgently requested by the aquaculture industry. Thus, the present review is timely and required by the industry and the scientific community as well. However, I have several concerns with the same as described below.

The structure of the paper is confusing. The introduction is extremely short, and the section presenting the mucosal immune response doesn’t make much sense. Instead of introducing one of the most important mechanisms used by the mucosal immune system to maintain homeostasis, the secretion of IgT achieved via de pIgR. This concept is lately approached between lines 309 – 311 but focused only on the IgM that is not the preferred immunoglobulin in the mucosa, thus not the target of immersion vaccination. More puzzling is that in the whole manuscript IgT is not mentioned at all. However, a discussion about the lymphoid tissue organization is presented in this early section. In my opinion, Irene Salinas already clearly described these issues in her 2015 paper (O-MALT vs D-MALT) https://www.ncbi.nlm.nih.gov/pmc/articles/PMC4588148/ 

Section 2 which is focused on salmonids doesn’t include trout despite it is mentioned in the same batch (Line 112). In this same section, the first 6 lines resemble more a short introduction, which should be fine if the following section will follow the same structure. However, is not the case. I think that the structure of each section should resemble the previous one. 

VER and VNN are treated as different pathogens but both are the same. 

For Sea bass, some references are not updated. The immersion vibrio vaccine mentioned, is the (in)famous Aquavac formerly announced as immersion oral vaccine as described in the text. However, after the effective oral demonstration by Galindo-Villegas and colleagues https://www.ncbi.nlm.nih.gov/pubmed/23932985 now it is sold exclusively as an oral preparation.

I don’t know which criteria used the present authors to include or not a fish species. However, due to the presence of grouper, guppy or gourami which are not extended species, I think that mandarin fish, pangasius, turbot, etc. which are relevant species for human consumption and results from a simple search in PubMed using immersion vaccination as query should be also included.

Several key references are missing. For example, since the late 90’s Nakanish and Otake already described must of the characteristics and constraints associated with the use of immersion vaccines presented in this MS https://www.ncbi.nlm.nih.gov/pubmed/9270835 

Huising from the group of J. Rombout did it so, but adding the novelty of the hyperosmotic pretreatment in common carp which is the most cultured fish in the world https://www.ncbi.nlm.nih.gov/pubmed/?term=PMID%3A+14505897 Unfortunately, in this MS only 4 lines out of 23 pages are dedicated to this species. 

More recently, Hoare and colleagues reported the efficacy of a polyvalent immersion vaccine in rainbow trout fry https://www.ncbi.nlm.nih.gov/pmc/articles/PMC5563058/ 

I do not doubt the expertise that both authors of this MS have on immunostimulants. However, the appreciation given to the same in this MS is extremely poor. 

On top of the above, I think that the narrative in most sections resembles a copy of what has been already published and lacks the constructive discussion that should characterize this kind of review on the aim of advancing the knowledge and not just repeating what we already know. This last appreciation is particularly applicable to the conclusion.

Author Response

Thanks for the evaluation of the manuscript. We are happy with the issues raised, and by undertaking a revision following the referees suggestions we are sure that the manuscript will be improved. Here is the point-by-point response:

"The structure of the paper is confusing. The introduction is extremely short, and the section presenting the mucosal immune response doesn’t make much sense. Instead of introducing one of the most important mechanisms used by the mucosal immune system to maintain homeostasis, the secretion of IgT achieved via de pIgR. This concept is lately approached between lines 309 – 311 but focused only on the IgM that is not the preferred immunoglobulin in the mucosa, thus not the target of immersion vaccination. More puzzling is that in the whole manuscript IgT is not mentioned at all. However, a discussion about the lymphoid tissue organization is presented in this early section. In my opinion, Irene Salinas already clearly described these issues in her 2015 paper (O-MALT vs D-MALT) https://www.ncbi.nlm.nih.gov/pmc/articles/PMC4588148/ " Response: We have elaborated more on the pIgR and IgT issue, with insertion of text and citations. Here is the added text: "The structure of the paper is confusing. The introduction is extremely short, and the section presenting the mucosal immune response doesn’t make much sense. Instead of introducing one of the most important mechanisms used by the mucosal immune system to maintain homeostasis, the secretion of IgT achieved via de pIgR. This concept is lately approached between lines 309 – 311 but focused only on the IgM that is not the preferred immunoglobulin in the mucosa, thus not the target of immersion vaccination. More puzzling is that in the whole manuscript IgT is not mentioned at all. However, a discussion about the lymphoid tissue organization is presented in this early section. In my opinion, Irene Salinas already clearly described these issues in her 2015 paper (O-MALT vs D-MALT) https://www.ncbi.nlm.nih.gov/pmc/articles/PMC4588148/" Response: We have added more on this topic with relevant citations. Here is the added text:  IgT (IgZ in zebra fish) has been proposed to be the central immunoglobulin produced at mucosal sites following exposure to parasites [54], and after oral vaccination with a IPNV DNA vaccine [55]. Furthermore, IgT expression was showed to be highly tissue-dependent after immersion vaccination against rainbow trout syndrome. In the latter study, IgT was temporarily increased in the posterior intestine while down regulated in gills and skin [56]. However, IgM is also present at mucosal sites after mucosal immunization, but generally not so regulated as IgT is. The relative importance of IgT response compared to IgM is debated. Inter-species differences, mode of vaccine administration, stimulation/immunization/infection may be decisive upon differential IgT and IgM responses [57]. Following on, not every teleost species possesses the IgT isotype [58]. It might appear that IgM is the only one Ig isotype in at mucosal sites since IgT response has been scarcely assessed after immersion vaccination in previous scholarly work. One reason for this is the lack of specific antibodies against IgT from various fish, which excludes immunohistochemical analysis and ELISA based assays. Local induction and production of both mucosal (IgT) and systemic (IgM) may be present upon microbial exposure [59], and in granulomatous tissues induced by intraperitoneal and intramuscular injection of vaccines [60,61]. Besides mucosal IgT, secretory polymeric Ig receptors (pIgR) with abilities to bind IgM and IgT has been proposed to transport IgM ang IgT to gut luminal area. A recent study showed increased skin pIgR and IgM mRNA expression following immersion vaccination in flounder (Paralichthys olivaceus). This finding was confirmed using ELISA and immunohistochemistry [62]. For more information on mucosal immunoglobulins and pIgR, see recent updates[58,63-67]. The expression of Ig and pIgR at mucosal sites is of vital importance with respect to mucosal binding and opsonization of IgM and IgT to antigens/pathogens". In addition, we have added a bit more information/text on lymphoid structures found in fish, as now the text is "It is acknowledged that the mucosal lining is of high importance to prevent pathogen entrance. Immune cells are present in all mucosal tissues/linings (MALT), from nasopharynx-associated lymphoid tissue (NALT), gill-associated lymphoid tissue (GIALT), skin-associated lymphoid tissue (SALT), buccal cavity associated lymphoid tissue, and to gut-associated lymphoid tissue (GALT). These tissues contain characteristics adaptive immune system [23-28]. The mucosal integument and soft tissues also contain innate defense molecules which may be fully protective against early stage infection [29]. These innate molecules may also be modulated by immersion treatment, feeding or by infection [29-50].
Strictly speaking, SALT and GALT do not contain well-organized lymphoid assemblages, but a more diffuse occurrence of immune cells – so-called diffuse mucosa-associated lymphoid tissue (D-MALT). Nevertheless, all “ALTs” may possess anti-infectious and pro-inflammatory elements important for protection, being innate lymphocyte derived mechanisms or immunoglobulins (in addition to innate defense factors). After immersion vaccination, there is a so-called disparity when it comes to antibody responses: Mucosal immunization induces particularly mucosal immune responses, whereas parenteral administration induces systemic response and production of specific antibodies [51-53]. It is difficult to pin-point which MALT tissue(s) that are most important when it comes to immune response and potential protection elicited after immersion vaccination. Likely, there are inter-species differences as well as differences between different mucosal sites, as pointed out by Khansari et al. 2018 [48]". "Section 2 which is focused on salmonids doesn’t include trout despite it is mentioned in the same batch (Line 112). In this same section, the first 6 lines resemble more a short introduction, which should be fine if the following section will follow the same structure. However, is not the case. I think that the structure of each section should resemble the previous one". Response: We have moved the chapter on rainbow trout to appear after the chapter "Atlantic salmon..", to make the order more logical. In addition, we have inserted more "general text" as "fish species introduction" in the first section below each fish species. This ease the reading a bit.

"VER and VNN are treated as different pathogens but both are the same". and "For Sea bass, some references are not updated. The immersion vibrio vaccine mentioned, is the (in)famous Aquavac formerly announced as immersion oral vaccine as described in the text. However, after the effective oral demonstration by Galindo-Villegas and colleagues https://www.ncbi.nlm.nih.gov/pubmed/23932985 now it is sold exclusively as an oral preparation". Response: We have fixed the VNN/VER issue. We have tried our best to update the citations for all species, including seas bass. Response: We have added the citation as suggested.

"I don’t know which criteria used the present authors to include or not a fish species. However, due to the presence of grouper, guppy or gourami which are not extended species, I think that mandarin fish, pangasius, turbot, etc. which are relevant species for human consumption and results from a simple search in PubMed using immersion vaccination as query should be also included". Response: We have added text to describe more fish species important for aquaculture industry, namely Mandarin fish, striped catfish, common carp, grass carp, and turbot. "Several key references are missing. For example, since the late 90’s Nakanish and Otake already described must of the characteristics and constraints associated with the use of immersion vaccines presented in this MS https://www.ncbi.nlm.nih.gov/pubmed/9270835". Response: Done. "Huising from the group of J. Rombout did it so, but adding the novelty of the hyperosmotic pretreatment in common carp which is the most cultured fish in the world https://www.ncbi.nlm.nih.gov/pubmed/?term=PMID%3A+14505897 Unfortunately, in this MS only 4 lines out of 23 pages are dedicated to this species". Response: We have included more text and citations. "More recently, Hoare and colleagues reported the efficacy of a polyvalent immersion vaccine in rainbow trout fry https://www.ncbi.nlm.nih.gov/pmc/articles/PMC5563058/" Response: This work had already been cited in the manuscript. "On top of the above, I think that the narrative in most sections resembles a copy of what has been already published and lacks the constructive discussion that should characterize this kind of review on the aim of advancing the knowledge and not just repeating what we already know. This last appreciation is particularly applicable to the conclusion". Response: We have added more recent work with citations, and tried our best to  constructively discuss (by adding our view) work published before. Since it is not our mission to make critique to previous work, but merely give an update of the field. As such, we think that we should´t  do a critical review on the topic covered. A critical review should also be made by the society. However, we have more or less completely rewritten our conclusion, and discussed our views in the introduction chapter. This will be in line with the referees suggestion.

Reviewer 2 Report

Title:

Review on immersion vaccines for fish: An update 2019

Recommendation:

Minor revision.

Comments:

This study updated and reviewed the immersion vaccines for fish. The topic of this review article is timing and interesting. Some questions and suggestions are offered below with the intent to assist the author in improving the manuscript.

In this study, one the reference mentioned about the poly(I:C), please provided more information about it’s characteristics and advantages while it applied in immersion vaccine. At line 220, author mentioned about “To minimize costs and maximize vaccine effectiveness without inducing immunological tolerance or suppression”. In previous study, does immersion vaccine have a case of immunological tolerance? Does the fish have a case of oral tolerance? Please provided more information about this part. Can author analyze the research breakthroughs in the improvement of immersion vaccine or the possible mechanism for the effects of new discoveries in recent years? At line 118, “Vaccines with 9.7%”, italic, typo? Page 7, at line 176-177 and 182, three terms, Intraperitoneal injection (IP), immersion (IMM) and anal intubation (AI) shown their abbreviations but the article does not follow these abbreviations. Page 9, line 283, there should be a comma between 75 and 11. Page 10, line 360, “NVV” might be a typo.

Author Response

Thanks for the review report witch addressed points to be addressed further and typo mistakes. As such, we have included/sorted out all points made by the referee in the revised manuscript.

"In this study, one the reference mentioned about the poly(I:C), please provided more information about it’s characteristics and advantages while it applied in immersion vaccine." Response: We have added more information on poly (I:C) with citation. "At line 220, author mentioned about “To minimize costs and maximize vaccine effectiveness without inducing immunological tolerance or suppression”. In previous study, does immersion vaccine have a case of immunological tolerance? Does the fish have a case of oral tolerance? Please provided more information about this part". Response: We have added citation (review) on this issue. "Can author analyze the research breakthroughs in the improvement of immersion vaccine or the possible mechanism for the effects of new discoveries in recent years?" Response: We have added "Other modalities to increase antigen uptake during immersion vaccination have been tried. The first one described increased adhesion and uptake of antigens of inactivated Flavobacterium when coated by positively charged chitosan which displayed mucoadhesive properties. This modality increased vaccine efficacy compared to what was obtained using naked vaccine antigens [17]. The other one used TNF alpha (TNF-a) nanoparticles which hold promise as an adjuvant for immersion vaccination[4]. Further on, resent studies pin point that nanoliposomes [18], recombinant live viruses expressing protective antigens and attenuated live vaccines [19-22], microbubbles[13], may be used to increase vaccine efficacy of immersion vaccines." in "introduction". "At line 118, “Vaccines with 9.7%”, italic, typo? Page 7, at line 176-177 and 182, three terms, Intraperitoneal injection (IP), immersion (IMM) and anal intubation (AI) shown their abbreviations but the article does not follow these abbreviations. Page 9, line 283, there should be a comma between 75 and 11. Page 10, line 360, “NVV” might be a typo." Response: Should be OK now.

Reviewer 3 Report

The manuscript: “Review on immersion vaccines for fish: An update 2019” (Microorganisms-ISSN 2076-2607) deals with a panorama of recent advances on vaccination protocols, as regards the immersion procedure for different commercial fish species in the world. The manuscript could be an interesting overview of the field but need some major revisions.

The English language needs a check, mainly in choosing words in the scientific form. Moreover, many points of contents (i.e, lacking the citation of species with high commercial interest, ignoring new procedures of vaccination, the possible use of mathematical models to obtain the best performance of larval vaccination; the probiotic use to enhance the performance of immunization), information, and scientific bibliography references are lacking. That is unacceptable for a manuscript with a “review” vocation of vaccination procedure panorama.

This manuscript, to be competitive as compared to other recently published papers, must be interested to give a piece of the correct and wide information. Some points are below detailed:

Introduction Lines 39-41: “Immersion vaccination comprises immersion of fish in water containing vaccine antigens. Dip vaccination is the most rapid method where the fish are immersed in water containing a relatively high dose of vaccine antigen(s) for one or several minutes”. The authors should re-write the sentences in only one, in the present form is too wordy. Lines 45-47: The authors ignore the manuscripts – Developmental & Comparative Immunology Volume 64, November 2016, Pages 118-137; Expert Review of Vaccines Volume 12, 2013 - Issue 5; Reviews in Fisheries Science & Aquaculture Volume 26, 2018 - Issue 1- Lines 50-56 The affirmations in the sentences, should including the concept that also a boosting of a vaccine in some critical time-points could be fundamental for the vaccination protocol next to the manuscripts cited in new dose duration hyperosmotic or protocols (Vaccine. 2013 Feb 6;31(8):1224-30). Moreover, further strategies in silico are now available to optimize the dose /duration of vaccine by the use of bioinformatics mathematical models (Bioinformatics, Volume 33, Issue 19, 01 October 2017, Pages 3065–3071). Mucosal immune response Lines 67-68 the authors should include some information about the full development of intestinal immune response (Cell Tissue Res (2007) 329:479–489) and the role of microbiota ( Fish Shellfish Immunol. 2013 Dec; 35(6): 1729–1739). Moreover. the use of probiotics could enhance strongly the immune response capacity in Fish (FSI 2014 Jul;39(1):78-89; FSI 2005, April 39(4) Pages 311-325; DCI Long-lived effects of administering β-glucans: Indications for trained immunity in fish -Volume 64, November 2016, Pages 93-102 Table 1 lack of some information regards Sparus aurata, a fish species massive aquacultured in Mediterranean area together with sea bass: Fish & Shellfish Immunology 16 (2004) 65–70; Aquaculture Volume 120, Issues 3–4, 1 March 1994, Pages 201-208; Fish & shellfish immunology 2005; Fish Shellfish Immunol.2015 Oct;46(2):292-6; J Immunol Res. 2014;2014:793817. doi: 10.1155/2014/793817. Moreover, the table and the manuscript do not include also Solea Solea, even aquacultured in Spain. Information on the use of nanoparticles delivery method: Reviews in Fisheries Science & Aquaculture Volume 26, 2018; https://doi.org/10.1080/23308249.2017.1334625 Suggestions on the species: Lines 200-274 the vaccination of channel catfish. The authors should consider J Aquat Anim Health. 2015 Jun;27(2):135-43. doi: 10.1080/08997659.2015.1032440; JWAS Volume47, Issue2 April 2016, Pages 207-211; Journal of the World Aquaculture Society 49(3) DOI: 10.1111/jwas.12515. also read in the web site: https://thefishsite.com/articles/oral-vaccine-could-save-us-catfish-farmers-millions; https://www.fis.com/fis/worldnews/worldnews.asp?monthyear=2-2019&day=20&id=101639&l=e&country=0&special=&ndb=1&df=0 Lines 406-429– vaccination of sea bass. The authors should consider a manuscript deals of herpes virus in sea bass: Virol J. 2019; 16: 71. Moreover, the authors should consider also the results of double pathogens (vibriosis/pasteurellosis) vaccination in the manuscript: Bioinformatics, Volume 33, Issue 19, 01 October 2017, Pages 3065–3071. Insert information on vaccination in Solea solea (Soil) and Sparus aurata (sea bream) fish species.

4) The conclusion should be re-write in view of the new references the should be inserted.

Author Response

The authors thanks for the referee´s suggestions to improve the manuscript. Please see the following rebuttals.

"The English language needs a check, mainly in choosing words in the scientific form. Moreover, many points of contents (i.e, lacking the citation of species with high commercial interest, ignoring new procedures of vaccination, the possible use of mathematical models to obtain the best performance of larval vaccination; the probiotic use to enhance the performance of immunization), information, and scientific bibliography references are lacking. That is unacceptable for a manuscript with a “review” vocation of vaccination procedure panorama". Response: We have done a language check and done copy editing to improve the english grammar and spelling. We have also added a lot more citations and added text on the suggested additional fish species, and also on new vaccination procedures. We have added text and citation on the use of mathematical modelling to depict vaccine performance. However, we do not find the use of probiotics to fit the current manuscript since it will add a lot of extra work, is quite outside the scope and could worsen the readability of the review. The other comments made by the referee have thoroughly been considered. While we have added text and references to most of the addressed issues, we have omitted suggested citations that fall outside the scope of the review. As example: The suggested references that concern oral vaccination have not been included. Oral vaccination by feed may be considered as mucosal immunization, but not immersion vaccination. This applies also to other work on parenteral immunisations. We have included more species in the table as suggested. We have rewritten the conclusion. We hope that the review is improved to be acceptable for publication. Please see attachment in which we have used track changes.

Round 2

Reviewer 1 Report

The fixed MS fairly improved the last version. Despite I don't fully agree in every concept exposed or the way they are presented, at this point, I could endorse the same once the authors add the following cites missing as previously suggested.

https://www.ncbi.nlm.nih.gov/pubmed/22790793

https://www.ncbi.nlm.nih.gov/pubmed/23932985 

I read in the response to R3 mentioning that oral administration is not the target of the present MS. Directly, it is not. However, to support some of the statements provided it would be desirable for the reader to have parallel contrasting information. Actually, some references exclusively focused on oral vaccination Gosh et al, 2016, Ballesteros et al., 2013, Mutoloki et al., 2015, etc. support the MS.

Please, change nonspecific and specific to innate and adaptive in L 253.

Author Response

Thanks for the comments and suggestions.

We have inserted one of the suggested reference (rTNFa as oral adjuvant), but not the other one since it is on oral vaccination exclusively. However, we have tried to conceptualise the main differences between oral and immersion vaccination - in the introduction (line 82-94) with relevant citations/work to underpin the differences. Ghosh, Mutoloki, Embregts have all been included to conceptualise this difference. The work by Ballesteros [60] was included as a reference to describe the mucosal immune system, not as a reference to oral vaccination. We acknowledge that this could mislead the reader. We have simply deleted this reference together with text (half a line).  We have changed "nonspecific and "specific" to "innate" and "adaptive"

Reviewer 3 Report

R2

The manuscript Microorganisms-ISSN 2076-2607 V2 was improved in some parts, however, it is still confusing in the text as respect the focus expressed by the title.

The title is: “Review on immersion vaccines for fish: An update 2019”

Postulated that “Vaccine may be administered to fish by three different routes: parenterally by injection, or by immersion (delivering the antigens to the fish surface by immersion of fish in a vaccine solution, or spraying the vaccine onto the fish), or orally by incorporating the vaccine into the feed”. This review should be focused on the second point, thus it should be focused on delivery antigens in immersion or spraying in/onto fish. I can agree with the author that oral vaccination/probiotics can be out of the aim, but of course also the DNA vaccination it is. Thus, is confusing the mention in the text when is treated as something new. 

I can imagine that in writing a review and in reading so many articles that all seem to be important and to cite in, but if the title is “immersion” the authors should focus on that, eventually, they have to change the title!

The focus, in my opinion, could be: 

Introduction: 

a panorama of the old typology of immersion procedure applied, and if the authors repute important to give information, a mention of other typology of vaccine delivery (i.e. injection, DNA vaccination, oral vaccination, microencapsulation, probiotics, and adjuvants). A short panorama of new kind of immersion procedure for fish that growth less fast than salmons of Atlantic cod (i.e. use of mathematical models; combination among vaccines/adjuvants/probiotics; different route of vaccination to stabilize the effectiveness of vaccination and combination with injection- for instance for reproductive fishes). New procedures potentially applied as mucoadhesive vaccines by electrostatic coating /chitosan as a possible future enhancement of all the new procedures. How the antigen can be processed by immersion procedure (i.e. gills, skin- uptake of the antigens)- the section called by authors:” mucosal immune response” Species by species the vaccination by immersion chosen by depending on the fast growth and the immune response.

The authors partially followed this scheme, in my opinion, they have to read again and make some changes. In the last version still, many typos are present (words attached, spaces missing).

Author Response

Thanks for the comments and suggestions.